# Digenic inheritance involving a muscle-specific protein kinase and the giant titin protein causes a skeletal muscle myopathy

In digenic inheritance, pathogenic variants in two genes must be inherited together to cause disease. Only very few examples of digenic inheritance have been described in the neuromuscular disease field. Here we show that predicted deleterious variants in *SRPK3*, encoding the X-linked serine/arginine protein kinase 3, lead to a progressive early onset skeletal muscle myopathy only when in combination with heterozygous variants in the *TTN* gene. The co-occurrence of predicted deleterious *SRPK3*/*TTN* variants was not seen among 76,702 healthy male individuals, and statistical modeling strongly supported digenic inheritance as the best-fitting model. Furthermore, double-mutant zebrafish (*srpk3*$^{-/-}$; *ttn.1*$^{+/-}$) replicated the myopathic phenotype and showed myofibrillar disorganization. Transcriptome data suggest that the interaction of *srpk3* and *ttn.1* in zebrafish occurs at a post-transcriptional level. We propose that digenic inheritance of deleterious changes impacting both the protein kinase SRPK3 and the giant muscle protein titin causes a skeletal myopathy and might serve as a model for other genetic diseases.

Over the past decade, next-generation sequencing (NGS) has contributed greatly to diagnostics in rare diseases. Nevertheless, many individuals considered to be affected by a genetic condition remain undiagnosed[1]. One underlying reason for this may be that the prevailing diagnostic paradigm still adheres to the one-gene one-disease model. Although thousands of monogenic diseases have been described, true digenic inheritance, where deleterious variants in two independent genes must be present for the disease to manifest, is scarce[2]. Here we describe a cohort of individuals with a skeletal muscle myopathy (henceforth referred to as myopathy) caused by co-inheritance of deleterious variants that impact both the muscle-specific protein kinase serine/arginine protein kinase 3 (SRPK3) and the giant muscle protein titin.

Titin, encoded by *TTN*, is the largest known protein and is expressed in cardiac and skeletal muscles. Spanning the Z-disk to the M-band, it is involved in sarcomeric assembly and function[3]. Expression and processing of *TTN* is age- and tissue-specific and involves complex transcriptional regulation[4,5]. Pathogenic variants in *TTN* cause a range of skeletal and cardiac phenotypes, which are inherited mostly in recessive[6–10] and dominant forms[11], respectively. However, due to its sheer size and extensive alternative splicing, interrogation and interpretation of genetic variants and protein expression data is challenging[12].

*SRPK3* encodes a protein kinase member of the SRPK family that phosphorylates proteins containing serine-arginine dipeptide motifs (SR proteins)[13]. In humans, three tissue-specific SRPKs have been described[14,15], with SRPK3 being expressed predominantly in striated muscle[16]. SRPKs primarily regulate both constitutive and alternative mRNA splicing through the phosphorylation of SR-splicing factors and spliceosomal components[13,17]. SRPK3 is essential for muscle growth

✉e-mail: ana.topf@ncl.ac.uk; volker.straub@ncl.ac.uk

and homeostasis[13,14,16], and skeletal muscle is highly sensitive to SRPK3 expression levels, as not only its deficiency but also its overexpression led to an abnormal muscle phenotype in mice, reminiscent of a human centronuclear myopathy[16]. In view of the mouse data, we initially considered *SRPK3* to be a myopathy candidate gene, but our subsequent findings support a more complex model.

## Results

### *SRPK3* variants alone do not explain disease manifestation

In our cohort of 2,170 exome datasets from patients with neuromuscular disease, we identified five males (of 1,170) hemizygous for deleterious variants in the X-linked *SRPK3* gene. Through international collaborations, we expanded the collection to a total of 33 patients with myopathy (31 males and two females, from 25 families) carrying deleterious variants in *SRPK3*. The majority (64%) were high-impact variants (stop gain, frameshift and splicing; Table 1). RNA analysis of splice variants showed abnormal mRNA splicing (Extended Data Fig. 1), expected to lead to nonsense-mediated decay, and *SRPK3* mRNA counts per million (CPM) in three individuals with truncating variants was significantly lower than controls ($-1.981$ log fold change, $P = 7.106 \times 10^{-11}$; Extended Data Fig. 2). All missense variants were located in one of SRPK3's kinase domains (Extended Data Fig. 3a) and were predicted to be deleterious (Table 1). 3D modeling anticipated restriction of the backbone conformation or disruption of the helical structure, causing instability and reduced catalytic activity (Extended Data Fig. 3b). All but one of the *SRPK3* variants were absent among 76,702 control males from gnomAD (v2.1.1; https://gnomad.broadinstitute.org/)[18]. The two manifesting female carriers in family Y (YIII:4 and YIII:5) showed skewed X-inactivation in lymphocytes, which could explain their phenotypes. Segregation analyses (in 20 families) showed that *SRPK3* variants were inherited from an unaffected mother, with none being de novo, and were present in all affected male siblings. However, *SRPK3* variants did not cosegregate with the disease as nine unaffected males from seven families (KII:2, LII:2, QII:1, VII:2/3/4, WIII:1, YIV:1 and ZIII:2) also carried the familial *SRPK3* variants (Fig. 1a and Supplementary Table 1).

We then noted that, in two of our extended *SRPK3* pedigrees (families M and Z; Fig. 1b), some family members presented with isolated dilated cardiomyopathy (DCM). In both families, the DCM was associated with a dominantly inherited heterozygous truncating variant in *TTN*—a new frameshift variant in exon 326 (p.Asp28805Metfs*6) in family M, and a stop-gain variant (p.Arg16095*), previously reported in association with DCM[19], in family Z. In these families, the patients with myopathy also presented with DCM and carried the familial 'cardiac' *TTN* variant. Interestingly, the myopathic phenotype (in patients MIII:3, ZIII:4, ZIII:7 and ZIV:1) only manifested when the *SRPK3* variant was present in combination with the *TTN* variant (Fig. 1b and Supplementary Table 1).

Taking this into account, we reassessed our myopathy families and screened them for *TTN* variants. All the index cases, in addition to the *SRPK3* variants, carried a heterozygous variant in the *TTN* gene. The vast majority (84%) were *TTN* truncating variants (*TTN*tv), and all were absent, or extremely rare, in the control population (Table 1). No variant clustering was observed (Extended Data Fig. 4). Segregation analyses showed that the myopathy manifested only if both the *SRPK3* and *TTN* variants were inherited together, but not when either variant was present in isolation (Fig. 1c). This was also the case for the two manifesting female carriers in family Y (YIII:4 and YIII:5), as they too carried a new deleterious heterozygous *TTN* variant. In contrast, females with cosegregating *SRPK3* and *TTN* variants but who had random X-inactivation were unaffected (that is, TI:2, UI:2, XII:2 and YII:2). The exception to this was RI:2, an unaffected 72-year-old female with cosegregating *SRPK3* and *TTN* variants whose fully inactivated chromosome (chr) X was confirmed to carry the *SRPK3* deleterious allele. Also unaffected was individual ZII:5, a female *SRPK3* carrier showing a fully skewed X-inactivation pattern (3:97) but no *TTN* variant (Fig. 1c and Supplementary Table 1).

### *SRPK3*/*TTN* cases present with a slowly progressive myopathy

Individuals with cosegregating *SPRK3*/*TTN* variants presented with a relatively homogenous phenotype and clinical course. Disease onset was in childhood or earlier (30/33), with poor motor performance. The disease was slowly progressive (23/33), yet all but four patients were ambulatory at the last assessment. The mean age of patients was 32 years (range: 1–77 years). The pattern of weakness was predominantly proximal and axial, affecting the lower more than the upper limbs. Respiratory compromise was present in 14 individuals, four of whom required noninvasive nocturnal ventilation. Three patients had DCM; however, this could be attributed to their *TTN*tv[11]. All patients had normal or mildly elevated serum creatine kinase (CK) levels, except RII:1, who had consistently elevated CK values (2,400 U/l). Less frequent features and deep-phenotype descriptions are listed in Supplementary Tables 2 and 3. Histopathology for 23 skeletal muscle biopsies showed myopathic changes with increased internalized nuclei (22/23), core-like structures (15/23) and type I fiber predominance (19/23). Electron microscopy (EM) images confirmed the presence of core-like areas and revealed myofibrillar disorganization with Z-line streaming and branching of myofibrils (Fig. 2). Axial T1-weighted muscle magnetic resonance imaging (MRI) scans of the lower limbs in four patients showed a similar selective pattern of muscle pathology with prominent fatty transformation of the subscapularis, gluteus maximus, adductor longus, vasti, hamstrings, medial gastrocnemius and soleus muscles. The sartorius, gracilis and adductor magnus muscles were well preserved (Fig. 2).

### Abnormal titin expression in patients with *SRPK3*/*TTN* myopathy

Based on the available genetic data, no evident copy number variants (CNVs) or variants in the triplicated region of the *TTN* gene were found in *trans* with the heterozygous *TTN* variant. However, to further exclude compound heterozygosity for a *TTN* defect abolishing the titin C-terminal, we used western blotting to detect the small C-terminal titin proteolytic fragments (13, 15 and 18 kDa in size)[20]. Muscle biopsy protein lysates of patients DII:1, LII:1, XIII:1 and YII:3 showed a normal titin C-terminal pattern (at least from one allele), ruling out a biallelic C-terminal titinopathy (Fig. 3a). Antibodies against the N-terminus and distal I-band of titin showed that the full-length titin band was present in a *TTN*tv carrier (DI:1) but appeared to be missing or reduced in the patients with *SRPK3*/*TTN* myopathy (CII:2, XIII:1 and YII:3), suggesting that the normal full-length expression of both *TTN* alleles was affected (Fig. 3b,c). In keeping with this, transcriptome analysis of patients LII:1, DII:1 and YII:3 showed that *TTN* mRNA expression (as length-normalized CPM) was significantly lower than controls ($-1.450$ log fold change, $P = 8.015 \times 10^{-4}$; Extended Data Fig. 5).

### *TTN* truncating variants are enriched in the *SRPK3* cohort

To elucidate whether the co-occurrence of *SRPK3* and *TTN* variants observed in our patients could be due to chance, we compared our findings with both control and other disease populations. We focused exclusively on *TTN*tv, as *TTN* missense variants are too abundant and would pose a challenge for correct pathogenicity ascertainment. First, we interrogated the gnomAD database and estimated that *TTN*tv were present at ~1% in the control population, in keeping with previous reports[21]. Next, we analyzed three cohorts of genetically confirmed limb-girdle muscular dystrophies: LGMD-R1 ($n = 170$), LGMD-R2 ($n = 94$) and LGMD-R12 ($n = 56$). While 21 of the 25 *SRPK3* families carried a *TTN*tv (84%), we found five heterozygous *TTN*tv in the patients with LGMD-R1 (2.94%; $P = 6.13 \times 10^{-19}$), one in the patients with LGMD-R2 (1.06%; $P = 9.18 \times 10^{-17}$) and two in the patients with LGMD-R12 (3.5%, $P = 4.89 \times 10^{-12}$; Extended Data Fig. 6). We then queried the existence of

**Table 1 | Genetic details of the patients with SRPK3/TTN myopathy**

| Fam | SRPK3 variants | | | | | TTN variants | | | | | | |
|---|---|---|---|---|---|---|---|---|---|---|---|---|
| | cDNA change | Protein change | Exon | Predicted effect (CADD score) | gnomAD freq. | cDNA change | Protein change | Exon | Band | Predicted effect (CADD score) | gnomAD freq. | Previously reported |
| A | c.1519+1G>A | p.? | in 14 | Splice-donor site | Absent | c.98810_98811del | p.Lys32937Argfs*5 | 354 | A-band | Frameshift | Absent | No |
| B | c.735dupC | p.Ser246Leufs*17 | 7 | Frameshift | Absent | c.93166C>T | p.Arg31056* | 340 | A-band | Stop gain | 1/248,360 | LOVD |
| C | c.1144+1G>A | p.Asp284_Thr383delinsAla | in 10 | Splice-donor site | Absent | c.95708G>A | p.Cys31903Tyr | 345 | A-band | Missense (24.6) | Absent | No |
| | | | | | | c.19234C>G | p.Pro6412Ala | 67 | I-band | Missense (20.5) | Absent | No |
| D | c.387+2_387+3delTG | p.? | 4 | Splice site | Absent | c.25480C>T | p.Arg8494* | 89 | I-band | Stop gain | Absent | Ref. [38] |
| E | c.475C>T | p.His159Tyr | 5 | Splice site | Absent | c.57168_57169insT | p.Ala19057Cysfs*6 | 294 | A-band | Frameshift | Absent | No |
| F | c.1301T>A | p.Val434Glu | 12 | Missense (24.6) | Absent | c.39226A>T | p.Lys13076* | 205 | I-band | Stop gain | Absent | No |
| G | c.1333G>A | p.Asp445Asn | 12 | Missense (25.1) | Absent | c.101440del | p.Glu33814Asnfs*7 | 358 | M-line | Frameshift | Absent | No |
| H | c.1289G>A | p.Arg430Gln | 12 | Missense (25.7) | Absent | c.38919del | p.Leu12974Trpfs*104 | 201[a] | N/a | Frameshift | 1/31,156 | No |
| J | c.388-2A>G | p.? | in 4 | Splice-acceptor site | Absent | c.37017del | p.Lys12339Asnfs*608 | 178[a] | N/a | Frameshift | Absent | No |
| K | c.1657C>T | p.Arg553Trp | 15 | Missense (29.1) | 1/181,513[b] | c.66699T>G | p.Tyr22233* | 317 | A-band | Stop gain | Absent | No |
| L | c.190+2T>C | p.? | 2 | Donor splice site | Absent | c.24019C>T | p.Arg8007* | 84 | I-band | Stop gain | Absent | No |
| M | c.1213_1218del | p.Lys405_Ile406del | 11 | Inframe | Absent | c.86413_86416delinsATG | p.Asp28805Metfs*6 | 326 | A-band | Frameshift | Absent | No |
| N | c.260G>A | p.Trp87* | 3 | Stop gain | Absent | c.95008C>T | p.Arg31670* | 342 | A-band | Stop gain | 1/248,798 | No |
| O | c.1070_1073del | p.Phe358Leufs*24 | 10 | Frameshift | Absent | c.103420C>T | p.Gln34474* | 358 | M-line | Stop gain | Absent | No |
| P | c.749-2A>G | p.? | in 7 | Splice-acceptor site | Absent | c.104092del | p.Arg34698Glufs*49 | 358 | M-line | Frameshift | Absent | No |
| Q | c.1363G>A | p.Glu455Lys | 13 | Missense (28) | 1/178,034[b] | c.77610del | p.Thr25871Glnfs*16 | 326 | A-band | Frameshift | Absent | No |
| R | c.1236delC | p.Asn412Lysfs*24 | 11 | Frameshift | Absent | c.89766G>C | p.Lys29922Asn | 336 | A-band | Missense (22.9) | Absent | No |
| S | c.774+5G>C | p.? | in 8 | Splice site | Absent | c.91085_91088del | p.Glu30362Glyfs*28 | 336 | A-band | Frameshift | Absent | No |
| T | c.804_807del | p.Lys269Argfs*2 | 9 | Frameshift | Absent | c.104947C>T | p.Gln34983* | 358 | M-line | Stop gain | Absent | No |
| U | c.587T>C | p.Leu196Pro | 7 | Missense (23.1) | Absent | c.24897del | p.Glu8300Asnfs*22 | 87 | I-band | Frameshift | Absent | No |
| V | c.392G>C | p.Arg131Pro | 5 | Missense (25.4) | Absent | c.76821C>A | p.Asn25607Lys | 327 | A-band | Missense (21.3) | Absent | No |
| | c.404C>A | p.Pro135His | 5 | Missense (28.7) | Absent | c.53938G>C | p.Ala17980Pro | 281 | A-band | Missense (23) | Absent | No |
| W | c.749-2A>C | p.? | in 7 | Splice-acceptor site | Absent | c.106259_106271del | p.Pro35420Leufs*54 | 359 | M-line | Frameshift | Absent | No |
| X | c.469G>A | p.Gly157Arg | 5 | Missense (45) | Absent | c.24087del | p.Lys8030Asnfs*13 | 84 | I-band | Frameshift | Absent | No |
| Y | c.1245G>A | p.Trp415* | 11 | Stop gain | Absent | c.70289T>A | p.Val23430Asp | 327 | A-band | Missense (23.5) | Absent | No |
| Z | c.1035dupC | p.Ala346Argfs*37 | 10 | Frameshift | Absent | c.48283C>T | p.Arg16095* | 258 | A-band | Stop gain | 1/119,862 | Ref. [19] |

All variants are reported according to the HGVS recommendations (http://varnomen.hgvs.org/). Genomic coordinates are based on GRCh37/hg19 assembly. CADD scores were calculated for missense changes only. Freq. indicates frequency within the largest available control population (gnomAD, https://gnomad.broadinstitute.org/). SRPK3 variants are annotated based on ENSG00000184343.6, NM_014370_3, transcript ENST00000370101.3 and NP_055185.2. TTN variants are annotated based on NG_011618.3 or LRG 391 and inferred-complete transcript variant-IC (NM_001267550.1 or ENST00000589042.5) and NP_001254479.1. TTN exon numbering is the LRG numbering (Leiden Open Variation Database, http://www.LOVD.nl/TTN). TITINdb was used to map the TTN variants to titin domains (http://fraternalilab.kcl.ac.uk/TITINdb). [a]Meta-transcript only exons, thought to be highly expressed during fetal development[5]. No missense variants in exons 344 or 364, known to be associated with HMERF and TMD, both dominant skeletal titinopathies (OMIM 603689 and 600334, respectively) were found. CADD scores for missense variants predicted them to be among the 0.005–0.00003% most damaging variants in the genome. [b]The SRPK3 p.Glu455Lys variant is found once in the control population in a healthy female carrier, whereas the SRPK3 c.1657C>T; p.Arg553Trp variant is found in a healthy male, who on manual inspection does not carry a TTN truncating variant. HGVS, Human Genome Variation Society; in, intron.

healthy male individuals carrying high-impact variants in both *SRPK3* and *TTN* in the control population of exomes from gnomAD. As of September 2023, there are six hemizygous males (of 76,702) carrying five truncating variants in *SRPK3* canonical transcript (p.Ser30Ter, p.Lys139GlnfsTer10, c.475+1G>A, p.Arg373Ter and p.Lys516SerfsTer17; ENSG00000184343; https://gnomad.broadinstitute.org/gene). Manual interrogation of the exome data of these six males disclosed that none of them carried a *TTN*tv, highlighting that co-occurring truncating hemizygous *SRPK3* and heterozygous *TTN* variants, as seen in our patients, is a very unusual event ($P = 0.00232$; Supplementary Note). Finally, we used statistical modeling to quantify the degree to which our observations supported the co-inheritance of causal *SRPK3* and *TTN* variants as opposed to any other plausible explanation. The best-fitting model involves digenic inheritance, with both *SRPK3* and *TTN* variants required for disease manifestation. The likelihood of data is at least $10^{10}$ times greater than under any other model, including a model where just one gene (that is, *SRPK3* or *TTN*) is operating, but with reduced penetrance (Supplementary Note).

## Zebrafish double mutants replicate the *SRPK3/TTN* human myopathy

We next tested whether our observations for *SRPK3* and *TTN* could be replicated in an animal model using *srpk3* and *ttn* zebrafish mutant lines—the *srpk3*^sa18907 mutation causes aberrant splicing, leading to partial retention of intron 15 or loss of exon 15 (Extended Data Fig. 7); *ttn.1*^sa5562 is a premature stop codon in exon 19 of 214 of the *ttn.1* gene. Zebrafish have two paralogs for the human *TTN* gene, *ttn.1* and *ttn.2*, with *ttn.1* exclusively affecting skeletal muscle function[22] (Extended Data Fig. 8). We created double carrier zebrafish of *srpk3*^sa18907 and *ttn.1*^sa5562 (*srpk3*^+/sa18907; *ttn.1*^+/sa5562, henceforth referred to as *srpk3*^+/−; *ttn.1*^+/−) and used sibling in-crosses to produce offspring carrying all possible genotype combinations, including *srpk3*^−/−; *ttn.1*^+/− (henceforth referred to as double mutant). Zebrafish single (*srpk3*^+/−; *ttn.1*^+/−) and double (*srpk3*^+/−; *ttn.1*^+/−) heterozygous mutant larvae at 5 dpf (days postfertilization; Fig. 4b,f) had no phenotype and were indistinguishable from the wild-type (WT; Fig. 4a,e). Double-mutant zebrafish (*srpk3*^−/−; *ttn.1*^+/−) at first appeared to be morphologically normal (including the heart), but they were not able to fill the swim bladder and therefore did not survive to adulthood. Compared to WT fish (Fig. 4a,e), the muscle fiber structure was largely lost in the *ttn*-null fish (both *srpk3*^+/+; *ttn.1*^−/− and *srpk3*^−/−; *ttn.1*^−/−; Extended Data Fig. 8), but was only very mildly affected in the *srpk3*-null fish (*srpk3*^−/−; *ttn.1*^+/+; Fig. 4c, g). In accordance with the findings in our patients with myopathy, however, the loss of one *ttn.1* WT allele in the *srpk3*-null zebrafish resulted in severe muscle pathology (*srpk3*^−/−; *ttn.1*^+/−; Fig. 4d,h), as visualized by whole mount staining of actin filaments and Z-band markers. Although myotomes were properly formed, muscle fibers were distorted and disintegrated to a variable extent. EM of the mutant zebrafish showed that the sarcomere appeared unaffected in heterozygous *ttn.1* (*srpk3*^+/+; *ttn.1*^+/−; Fig. 3r), comparable

to WT zebrafish (Fig. 4q). The *srpk3*-null zebrafish (*srpk3*^−/−; *ttn.1*^+/+; Fig. 4s) showed mildly disorganized myofibrils; however, most sarcomeres appeared well defined. The double-mutant fish (*srpk3*^−/−; *ttn.1*^+/−; Fig. 4t) displayed pronounced disruption of the sarcomere structure, including the myofibrils, A-band, I-band, H-zone and M-line. For further characterization, we isolated zebrafish myofibers and immunostained them with a monoclonal anti-titin antibody. This antibody labeled the T11 peptide found at the I-band to A-band transition. We observed, by confocal imaging, that *srpk3*^−/−; *ttn.1*^+/− double mutants (Fig. 4l,p) developed disorganized sarcomeres compared both to *ttn.1* heterozygous (*srpk3*^+/+; *ttn.1*^+/−; Fig. 4j,n) and *srpk3*-null mutants (*srpk3*^−/−; *ttn.1*^+/+; Fig. 4k,o). This disruption of the myofibers was also illustrated by a substantial reduction of titin labeling and disturbance of the transverse labeling pattern of α-actinin. These results were consistent with the I-band and A-band alterations observed in the EM images.

## Transcriptome analysis highlights disruption of contractile structures in zebrafish double mutants

Zebrafish transcriptome data showed that *ttn.1* mRNA expression is equally reduced in the heterozygous (*srpk3*^+/+; *ttn.1*^+/−) and the double mutants (*srpk3*^−/−; *ttn.1*^+/−) when compared to WT *ttn.1* (Extended Data Fig. 9a). This suggests that the severe reduction of titin protein expression observed by immunostaining exclusively in the double mutants (*srpk3*^−/−; *ttn.1*^+/−; Fig. 4l) was due to aberrant post-transcriptional or post-translational processing. In contrast, *srpk3* mRNA levels were upregulated in *srpk3*^−/− mutants (both *srpk3*^−/−; *ttn.1*^+/+ and *srpk3*^−/−; *ttn.1*^+/−; Extended Data Fig. 9b), likely as a compensatory effect. When analyzing the global transcriptome profile of the different genotypes, we observed that there were 128 genes differentially expressed (DE) in the heterozygous *ttn* (*srpk3*^+/+; *ttn.1*^+/−) zebrafish. The number of DE genes in the *srpk3*-null mutant (*srpk3*^−/−; *ttn.1*^+/+) was 572, and this increased to 794 in the double mutant (*srpk3*^−/−; *ttn.1*^+/−; Fig. 5a). A Gene Ontology (GO) enrichment analysis to identify the differential pathways involved showed that the transcriptional changes in *srpk3*-null (*srpk3*^−/−; *ttn.1*^+/+) zebrafish and the double mutant (*srpk3*^−/−; *ttn.1*^+/−) were similar. Expression of genes involved in myofibril and actin cytoskeletal organization and skeletal muscle tissue development was affected. There was also an inflammatory signal in both genotypes (Fig. 5b,c, Extended Data Fig. 10 and Supplementary Data). Unexpectedly, the heterozygous *ttn.1* mutant (*srpk3*^+/+; *ttn.1*^+/−), despite being fully viable and having no morphological phenotype, also showed a clear inflammatory signature, but no dysregulation of muscle genes (Fig. 4b and Supplementary Data). In addition, given the role of SRPK3 in RNA processing and maturation, we queried for generalized aberrant splicing patterns, but we did not detect any signal for this.

## *Ttn* variants in *Srpk3* knockout (KO) mouse

We then interrogated the genetic background of the *Srpk3* KO mouse described in ref. 16 to establish whether its resulting muscle phenotype

**Fig. 1 | Pedigrees of the *SRPK3/TTN* myopathy families. a**, Segregation of the familial *SRPK3* variants is shown. S indicates the *SRPK3* variant and WT indicates the wild-type allele. Individuals presenting with skeletal muscle disease are indicated in black. Mild presentations are shown in gray (corresponding to YIII:4 and YIII:5, two female carriers with skewed X-inactivation, 80:20 and 65:35, in lymphocytes, respectively). **b**, Extended pedigree details of families M and Z. Individuals presenting with skeletal muscle disease are indicated in black. Cardiac involvement is indicated by gray/dotted symbols. Segregation of the familial *SRPK3* (S) and *TTN* (T) variants is shown. S + T indicates cosegregating *SRPK3/TTN* variants; WT indicates both *SRPK3* and *TTN* WT alleles. Individuals ZIV:1, ZIV:4, ZIV:6 and ZIV:7 carry the familial *TTN* variant (p.Arg16905*) previously reported in association with DCM (ref. 19) but are presymptomatic at ages 52, 44, 40 and 38 years old, respectively. Likewise, individual MIII:2 carries the familial *TTN* variant (p.Asp28805Metfs*6) but is also presymptomatic at

age 46 years old. **c**, Cosegregation of the *SRPK3* and *TTN* variants (S + T) with the myopathic phenotype (shown in black). All known genotypes are shown; WT, both *SRPK3* and *TTN* WT alleles; empty symbols indicate that the sample was not available for testing (or failed testing). All affected individuals carry the *SRPK3* and *TTN* variants (S + T), whereas their unaffected relatives carry one or the other, but never both. Two females carrying cosegregating *SRPK3/TTN* variants and showing a skewed X-inactivation pattern are mildly affected (YIII:4 and YIII:5), and those with random X-inactivation are unaffected (TI:2, UI:2, XII:2 and YII:2). A female carrying only the *SRPK3* variant (but no *TTN* variant; ZII:5) and a complete X-inactivation pattern (3:97, in lymphocytes) is unaffected. Individual RI:2, with cosegregating *SRPK3* and *TTN* variants whose fully inactivated chr X carries the *SRPK3* deleterious variant, is also unaffected. Individuals RII:3 and SI:2 are noninformative for the CAG repeat analyzed in the X-inactivation assay.

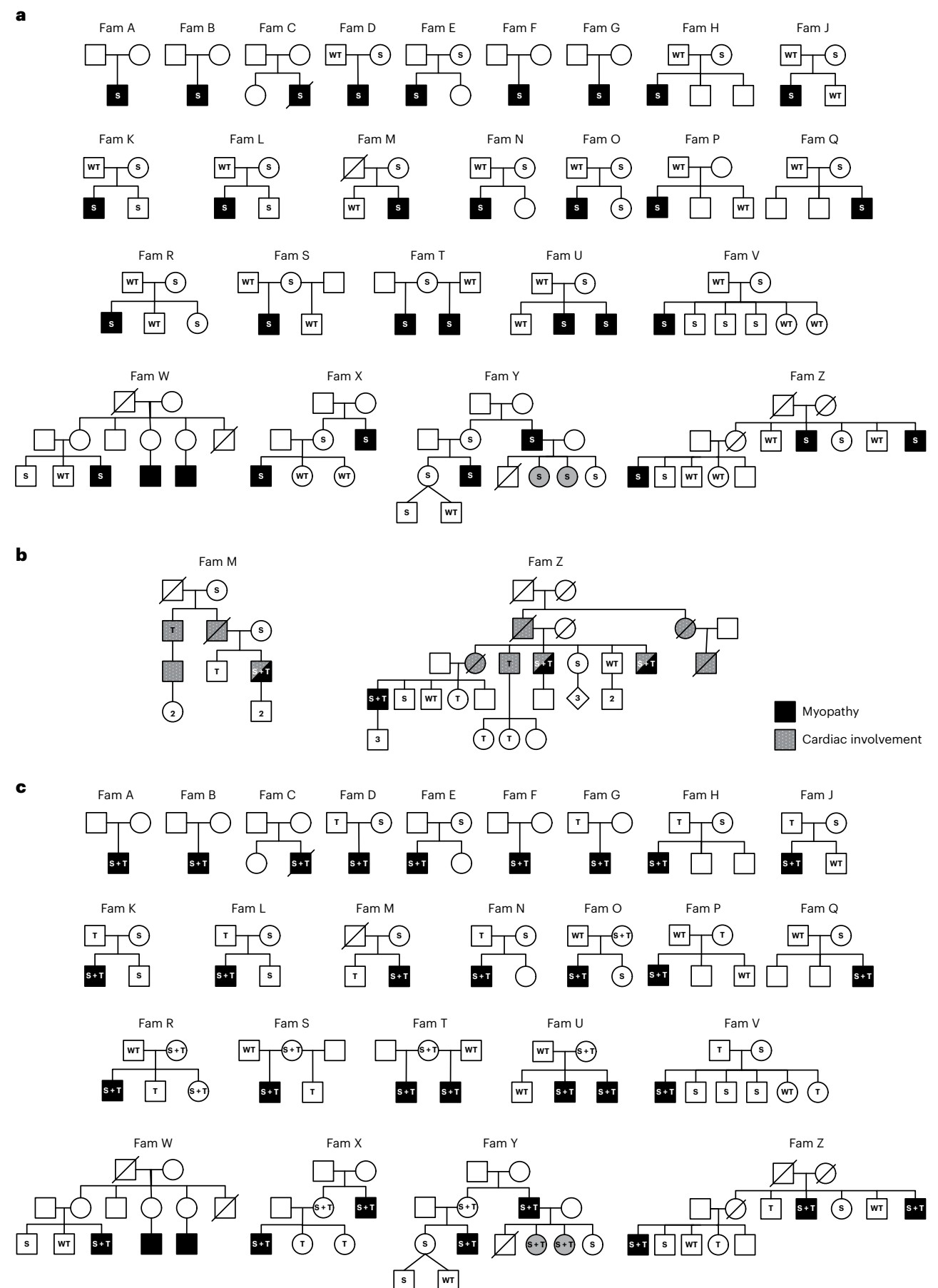

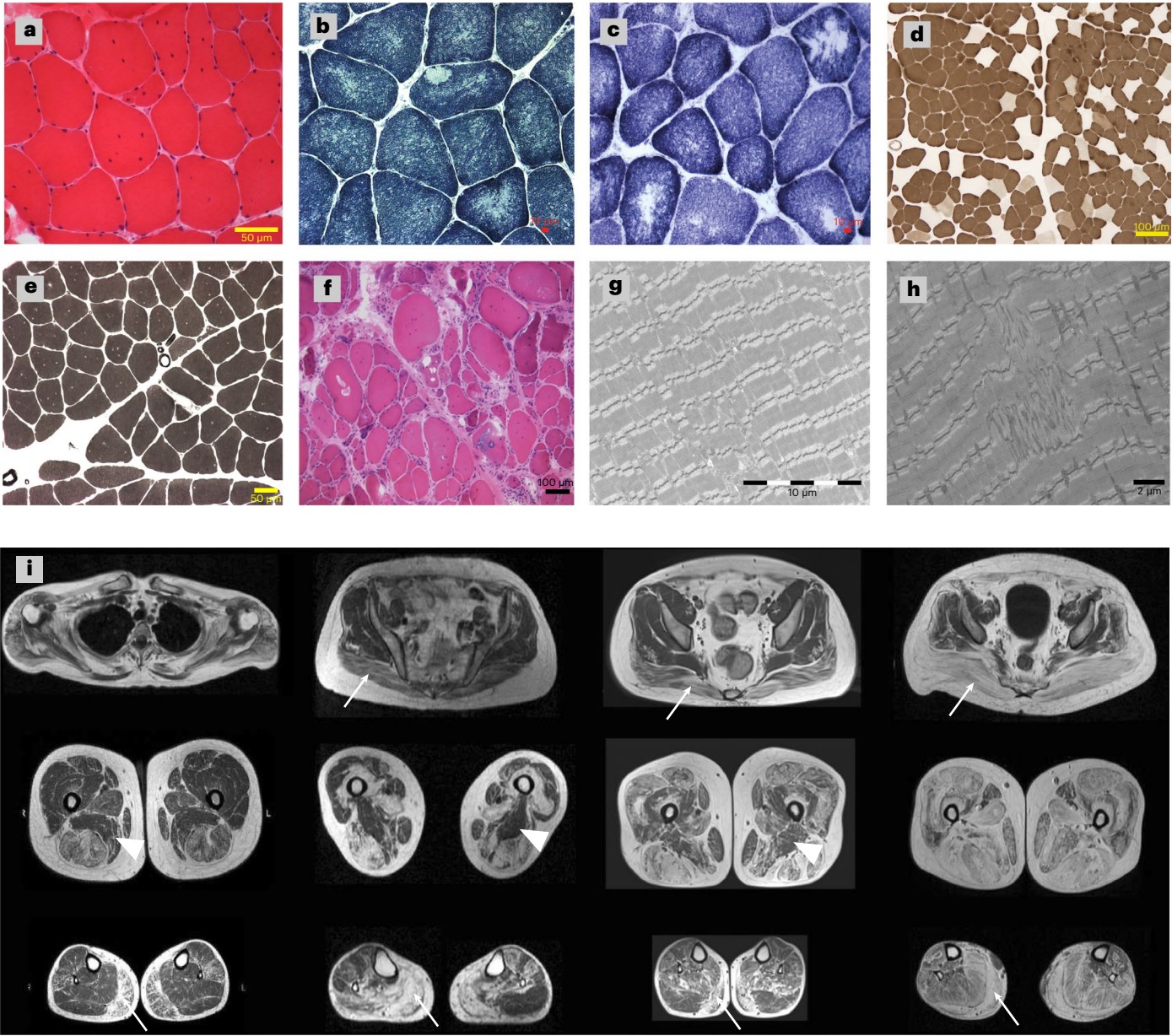

**Fig. 2 | Muscle pathology of the patients with *SRPK3*/*TTN* myopathy.**
**a–h**, Examples of muscle histopathology (*n* = 23). **a**, Myopathic changes
with increased internalized nuclei and fiber size variability (22/23) shown by
hematoxylin and eosin (H&E) staining, as seen in patient XIII:1. **b**,**c**, Minicores
and core-like structures (15/23) shown by NADH histochemistry, as seen in
patients BII:1 and WIII:3. **d**,**e**, Type I fiber predominance and type I uniformity
(19/23) shown by ATPase pH 4.6 and pH 9.2 staining, as seen in patients CII:2 and
BII:1, respectively. **f**, More severe end of the disease spectrum, with vacuoles,
necrosis, regeneration and fibrosis shown by H&E in patient YII:3. **g**,**h**, EM images
confirmed the presence of core structures and revealed Z-line misalignment,
accumulation of Z-band material and branching of myofibrils, as seen in patients
XIII:1 and WIII:3. Representative images have been obtained as part of the
diagnostic workup in accredited pathology laboratories. **i**, Lower limb MRI T1-
weighted images from four patients with *SRPK3*/*TTN* myopathy (VII:1, YII:3, MIII:3
and ZIV:1). A pattern of fatty replacement involving the subscapularis muscle in
the shoulder girdle was observed. In the pelvic girdle, the gluteus maximus was
affected (arrows), but the gluteus minimus and medius muscles were spared
even in the advanced stages of the disease. In the thigh, there was a predominant
involvement of the hamstring muscles, while the sartorius and gracilis muscles
were not involved in the advanced stages of the disease, with the adductor
magnus muscle (arrowheads) almost completely spared. In the lower legs, there
was predominant involvement of the medial gastrocnemius muscle (arrows)
associated with the involvement of the soleus muscle. The peroneus and tibialis
anterior muscles were also involved, but only in advanced stages.

was also due to the cosegregation of *Srpk3* and a previously unre-
vealed *Ttn* variant. Using genome sequencing, we found the follow-
ing three *Ttn* changes in the *Srpk3* KO mouse model: two missense
(chr2:76946873C>T; p.Ala1395Val and chr2:76969682C>T; p.Ala394Val)
and one synonymous variant (chr2:76969699T>C; p.Ser388Ser), also
present in the WT 129s6/SvEvTAC background (http://www.informat
ics.jax.org/snp/). We currently do not know whether these variants
contribute to the observed phenotype.

## SRPK3 in vitro phosphorylates RNA-binding motif 20 (RBM20), a splicing factor involved in *TTN* mRNA regulation

Protein expression analysis of muscle biopsy lysates from our patients
with *SRPK3*/*TTN* myopathy suggested that SRPK3 deficiency might
affect normal full-length titin expression. This was later supported
by the reduction in titin labeling seen in the zebrafish double-mutant
(*srpk3*[−/−]; *ttn.1*[+/−]) model. SRPK3 could be directly involved in titin phos-
phorylation or more likely, given the regulatory role of serine/arginine

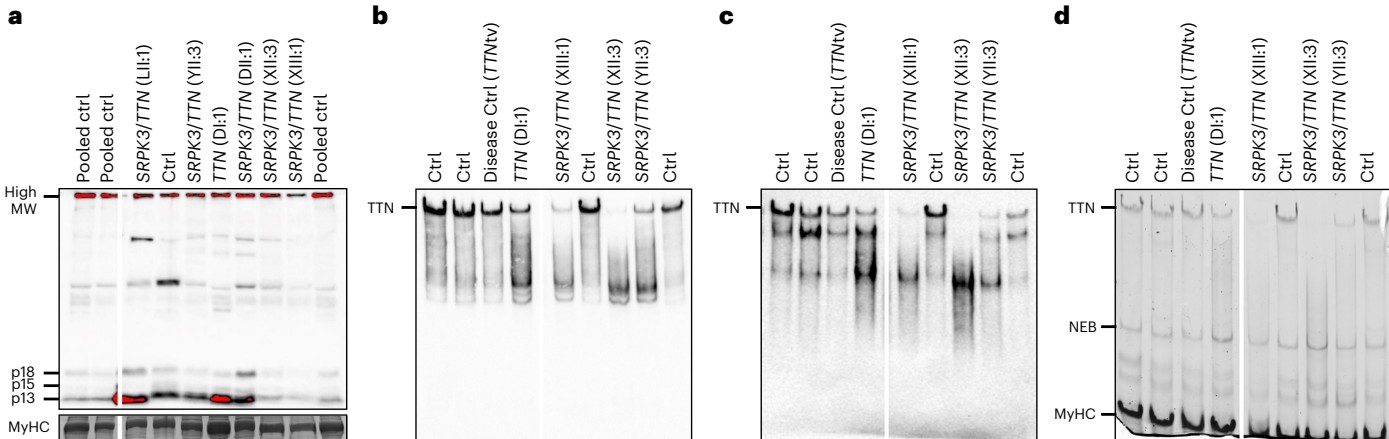

**Fig. 3 | Titin immunoanalysis of patients with *SRPK3*/*TTN* myopathy.** Muscle biopsy lysates of individuals DI:1, DII:1, LII:1, XII:3, XIII:1 and YII:3 were analyzed using different anti-titin antibodies. **a**, *SRPK3*/*TTN* patients LII:1, YII:3, DII:1, XII:3 and XIII:1 showed a normal pattern of C-terminal titin proteolytic fragments (13, 15 and 18 kDa in size), ruling out a C-terminal titinopathy. **b**,**c**, Antibodies against the N-terminal titin (Z1Z2 TTN-1, **b**) and distal I-band of titin (F146.9B9, **c**) showed that the full-length titin band is missing or highly reduced in the patients with *SRPK3*/*TTN* myopathy (XIII:1, XII:3 and YII:3), but it is present in an unaffected

relative *TTN*tv carrier (DI:1, father of DII:1) and a disease control also carrying a heterozygous *TTN*tv. This could be attributed to changes in N-terminal protein sequence or structure, or otherwise, protein modifications preventing antibody recognition. **d**, Coomassie staining also showed the absence or reduction of the high molecular weight band representing the full-length titin protein, whereas the NEB and MyHC bands were normal. Western blots were repeated twice, from the same muscle lysates. Full-length blots are provided as source data. MW, molecular weight; MyHC, myosin heavy chain; NEB, nebulin.

(SR) kinases, in *TTN* mRNA processing by targeting an SR-splicing regulator. RBM20 is a muscle-specific splicing factor involved in *TTN* alternative splicing[23]. Its cellular localization and activity are dependent on the phosphorylation of its RSRSP stretch within the arginine/serine-rich region[24]. We hypothesized that RBM20 might be a phosphorylation substrate of SRPK3 and the mediating link between SRPK3 and titin, as previously shown for SRPK1 (ref. [25]). To investigate this, we cotransfected an RBM20 reporter (RBM20$_{517-664}$-V5) into 293T cells with or without a GFP-SRPK3 construct. The presence of GFP-SRPK3 led to RBM20$_{517-664}$-V5 hyperphosphorylation, as indicated by a mobility shift of the reporter (Fig. 6a). This mobility shift was abolished after treatment with lambda phosphatase. This suggests that SRPK3 may directly phosphorylate the *TTN*-specific splicing factor, RBM20. Based on this finding, we interrogated the transcriptome data for the zebrafish mutants and found that the zebrafish *rbm20* ortholog shows increased expression upon loss of *srpk3* (both *srpk3*$^{-/-}$; *ttn.1*$^{+/+}$ and *srpk3*$^{-/-}$; *ttn.1*$^{+/-}$; Fig. 6b).

## Discussion

We identified 40 males (from 25 families) carrying hemizygous deleterious variants in the X-linked *SRPK3* gene. Of those, only the 31 patients who also carried cosegregating heterozygous *TTN* variants presented with a myopathy. Their unaffected brothers carried either the *SRPK3* or the *TTN* variant, but never both. For the female individuals, a mild presentation was observed only in the two sisters from family Y who carried both the *TTN* and the X-linked *SRPK3* variants and showed skewed X-inactivation. However, a female *SRPK3* carrier displaying skewed X-inactivation, but no *TTN* variant, was unaffected. The remaining females with *SRPK3*/*TTN* variants but random X-inactivation were also unaffected. While numbers are small, this might suggest that, for *SRPK3* carrier females to present a myopathic phenotype, both skewed X-inactivation and a deleterious *TTN* variant must co-occur, in line with what is observed in male patients.

Disease and control population data indicated that the co-occurrence of *SRPK3* and *TTN* variants was not fortuitous, because *TTN*tv variants were significantly more common in patients with *SRPK3*/*TTN* myopathy than in other genetically diagnosed muscular dystrophy cohorts, and were notably absent in the 'SRPK3-null' males present in the control population. These findings, together with the

statistical modeling, strongly support the digenic inheritance of deleterious *SRPK3* and *TTN* variants in patients with myopathy. While digenic inheritance has been widely recognized in association with, for example, deafness[26,27] and cardiovascular conditions[28], only a handful of digenic cases have been reported in the neuromuscular field. These are, however, mostly single cases[29,30], with only *SQSTM1*/*TIA1* multisystem proteinopathy (MPS)[31] and D4Z4/SMCHD1 facioscapulohumeral muscular dystrophy type 2 (FSHD2; ref. [32]) being replicated in independent cohorts. To our knowledge, this is the first report of true digenic inheritance involving a protein kinase in a sizeable cohort of skeletal myopathy patients.

The zebrafish model, where the *srpk3*$^{-/-}$; *ttn.1*$^{+/-}$ double-mutant embryos showed a severe muscle phenotype not observed in the *srpk3*$^{-/-}$ or *ttn.1*$^{+/-}$ embryos alone, replicated our findings. In addition, the model allowed us to better understand the muscle pathology, highlighting the disorganization of the sarcomere and the reduction in titin protein expression. Transcriptome analysis showed that compared to WT, *ttn.1* mRNA expression levels were similarly reduced in the single *ttn.1*$^{+/-}$ heterozygous mutant, both with or without *srpk3*-null background, suggesting that post-transcriptional (or post-translational) processing must be responsible for the abnormally expressed protein. Likewise, differential gene expression analysis demonstrated that, despite severe morphological consequences, losing one *ttn.1* WT allele in *srpk3*-null mutants only had a minor effect on the gene expression profile. This suggests that the interaction of *srpk3* and *ttn.1* in zebrafish is at a post-transcriptional level.

The in vitro phosphorylation assay supported a connection between SRPK3 and the *TTN*-splicing factor RBM20. Increased mRNA counts of the *rbm20* zebrafish ortholog were observed exclusively in the *srpk3*-null mutants, possibly the result of a positive feedback loop due to srpk3-related phosphorylation deficiency of rbm20. Similar upregulation was seen in a knock-in RBM20 mouse model (Rbm20$^{S637A}$) where phosphorylation was impaired[33]. While in our hands the *srpk3*-null 5-dpf zebrafish appeared normal with almost unaffected fiber structure, at the time of submission an adult KO model was shown to present agenesis of cerebellar structures and abnormal behavior[34], suggesting srpk3 may also be involved in neural development.

Most of the *TTN* variants identified in our *SRPK3* cohort were truncating, yet novel missense variants, predicted deleterious, were also

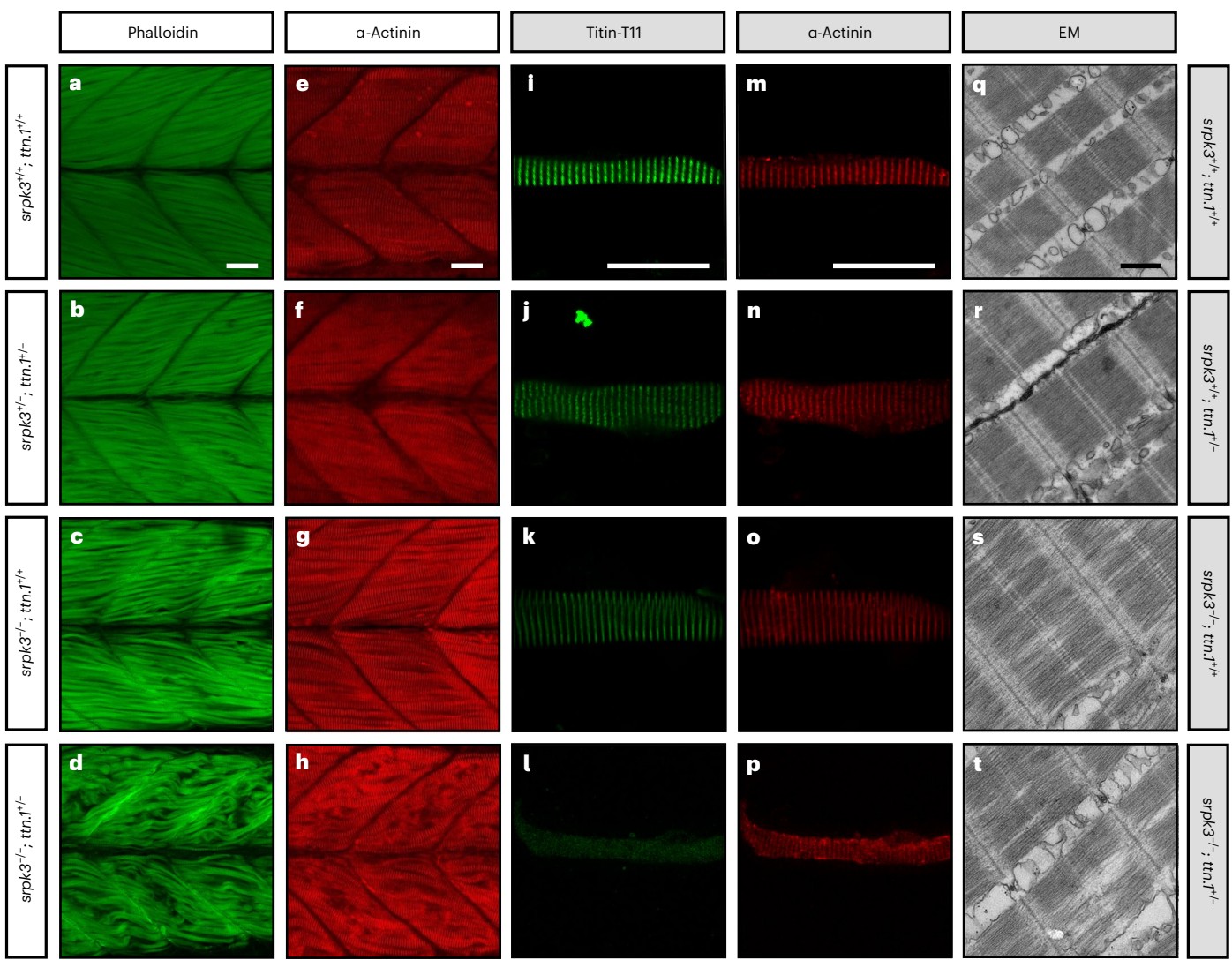

**Fig. 4 | *ttn.1* heterozygosity induces a severe phenotype in homozygous *srpk3*-null mutant zebrafish larvae. a–h**, Lateral view of Alexa Fluor phalloidin filamentous actin (green) and α-actinin Z-band marker (red) staining in skeletal fast muscle fibers in WT (**a,e**), *srpk3*⁺/⁺; *ttn.1*⁺/⁻ (**b,f**), *srpk3*⁻/⁻; *ttn.1*⁺/⁺ (**c,g**) and *srpk3*⁻/⁻; *ttn.1*⁺/⁻ larvae (**d,h**) at 5 dpf. Compared to WT (**a,e**) or double heterozygotes (*srpk3*⁺/⁻; *ttn.1*⁺/⁻; **b,f**), homozygous *srpk3*-null alone only causes very mild muscle fiber defects (**c,g**), while *ttn.1* heterozygosity in homozygous *srpk3*⁻/⁻ larvae severely affects muscle fiber integrity (**d,h**). **i–t**, Isolated myofiber immunostaining and electron microscopy (EM) in skeletal fast muscle fibers in WT (**i,m,q**), *srpk3*⁺/⁺; *ttn.1*⁺/⁻ (**j,n,r**), *srpk3*⁻/⁻; *ttn.1*⁺/⁺ (**k,o,s**) and *srpk3*⁻/⁻; *ttn.1*⁺/⁻ (**l,p,t**) larvae at 5 dpf. Isolated myofiber immunostaining showed that titin expression is largely reduced in the double mutant (*srpk3*⁻/⁻; *ttn.1*⁺/⁻; **l,p**) but not in the single heterozygous *ttn.1* mutant (*srpk3*⁺/⁺; *ttn.1*⁺/⁻; **j,n**) or the *srpk3*-null (*srpk3*⁻/⁻; *ttn.1*⁺/⁺; **k,o**). EM showed that *srpk3*-null zebrafish (*srpk3*⁻/⁻; *ttn.1*⁺/⁺; **s**) had well-defined sarcomeres, with mildly disorganized myofibrils. The double-mutant fish (*srpk3*⁻/⁻; *ttn.1*⁺/⁻; **t**) displayed pronounced disruption of the sarcomere structure. White scale bars are 25 μm. Black scale bar is 500 nm. Representative images from >15 pooled fish per genotype.

found. While more challenging to ascertain[35], *TTN* missense changes have been shown to be disease-causing in homozygosity or compound heterozygosity with a *TTN*tv or other missense change[35–37]. In heterozygosity, in particular in exons 344 and 364, *TTN* missense changes are associated with dominant hereditary myopathy with early respiratory failure (HMERF)[10,11] and tibial muscular dystrophy (TMD)[7,9]. Three of the *TTN*tv variants seen in our cohort were previously reported. Two of these (p.Arg16095* and p.Arg31056*) had been associated with DCM (ref. 19 and Leiden Open Variation Database, https://databases.lovd.nl/shared/genes/TTN) and were identified in two (of the three) patients with *SRPK3*/*TTN* myopathy also presenting with DCM (families B and Z). The third variant (p.Arg8494*) had been reported in a patient with an unsolved muscle disease[38] who was analyzed through a panel of 35 neuromuscular disease genes. Given our findings, it would be appropriate to screen the *SRPK3* gene in such unsolved patients with sporadic myopathy and a heterozygous *TTN*tv.

Notably, none of the family members who carried only the heterozygous *TTN* variant (*n* = 16) showed any signs of skeletal muscle disease, in keeping with what has been largely accepted for heterozygous *TTN*tv[6,8,20]. Notwithstanding, it has been reported recently that heterozygous *TTN*tv in the A-band may be causative of dominant distal myopathy[39]. When no evident dominant family history exists, however, it would be worth considering whether a more complex molecular pathomechanism might be responsible for these presentations.

N-terminal blots of a *TTN*tv carrier showed expression of full-length titin, yet when similar variants were present in combination with *SRPK3* variants, only smaller or weaker bands seemed to be detected. Similarly, reduction of titin immunolabeling was observed in the heterozygous *ttn.1* zebrafish model but only in an *srpk3*-null background. The epitope for the anti-titin antibody is located in region T11 (around exon 102), downstream of the premature stop codon generated by the *ttn.1*ˢᵃˢ⁵⁶² mutation (chr9:42861631T>G, exon 19); therefore, only

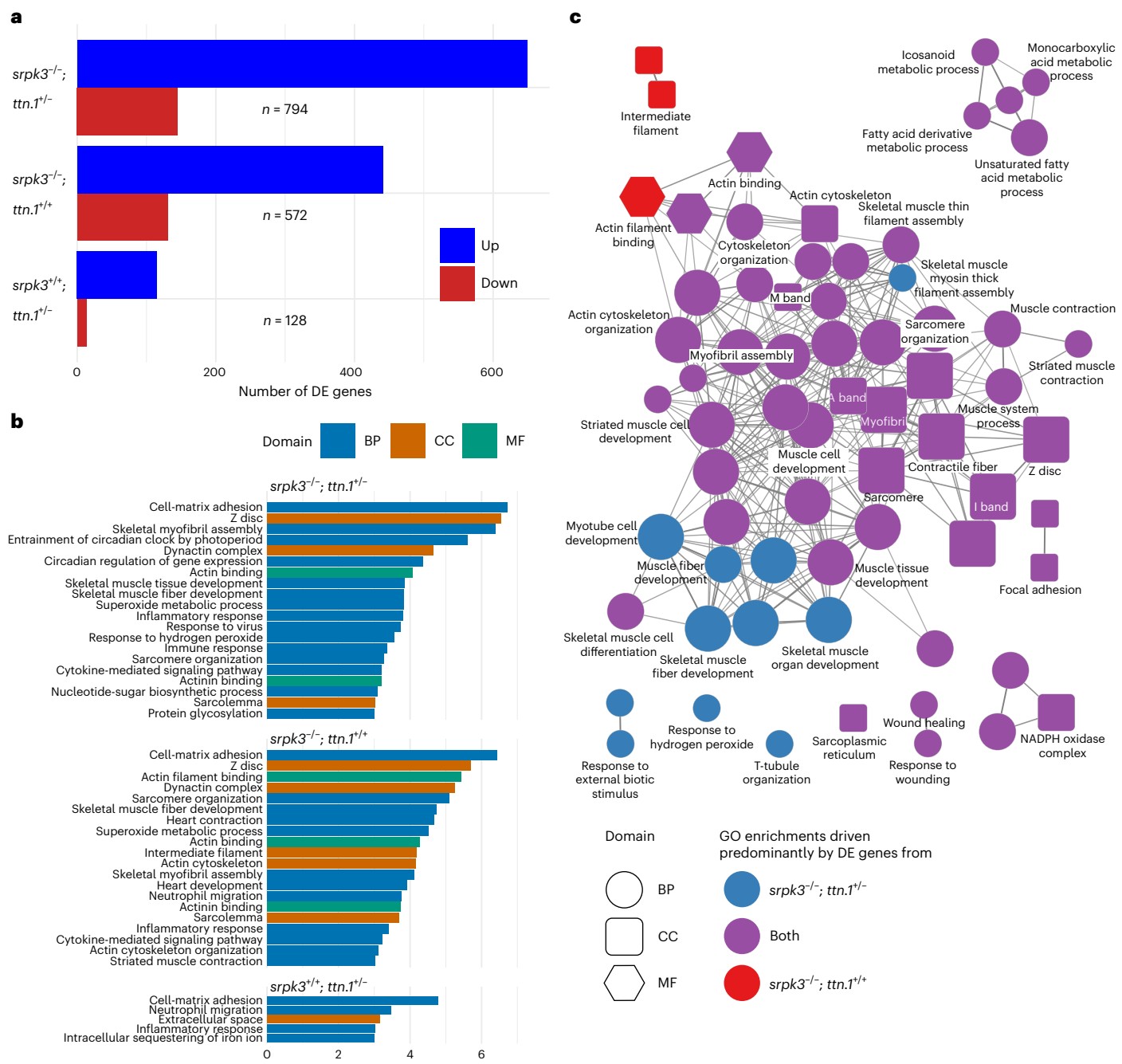

**Fig. 5 | Transcriptome analysis of mutant zebrafish larvae. a**, Number of DE genes between WT and double mutant (top: *srpk3⁻/⁻; ttn.1⁺/⁻*; *n* = 794), *srpk3*-null (middle: *srpk3⁻/⁻; ttn.1⁺/⁺*; *n* = 572) and heterozygous *ttn.1* (bottom: *srpk3⁺/⁺; ttn.1⁺/⁻*; *n* = 128) zebrafish. Upregulated genes are in blue and downregulated genes are in red. **b**, GO term enrichment analysis. GO term enrichment was done using the topGO package using a one-sided Fisher's exact test without adjustment for multiple testing. The top enriched GO terms (*P* < 0.001) for the three comparisons in **a** ordered by −log₁₀(*P*). The bars are colored according to the GO domain. Blue indicates BP; orange indicates CC; green indicates MF.

**c**, ClueGO network diagram showing the overlap in enriched GO terms between double mutant (*srpk3⁻/⁻; ttn.1⁺/⁻*) and *srpk3*-null (*srpk3⁻/⁻; ttn.1⁺/⁺*). Nodes represent individual enriched GO terms; edges connect nodes that share annotated genes from the DE genes. Nodes are colored according to the contribution to the enrichment from DE genes from each comparison. Blue indicates >60% DE genes from the *srpk3⁻/⁻; ttn.1⁺/⁻* comparison; red indicates >60% DE genes from *srpk3⁻/⁻; ttn.1⁺/⁺*; purple indicates 40–60% from each comparison. BP, biological process; CC, cellular component; MF, molecular function.

WT titin would have been detected by immunostaining. This suggests that the loss of SRPK3 negatively affects the WT *TTN* copy, either by directly altering protein structure or conformation, or more likely, by post-transcriptional processing (possibly through RBM20 regulation), resulting in loss of antibody recognition. We propose that the myopathy observed in the patients with *SRPK3/TTN* myopathy, and replicated in the zebrafish model, is the result of a titin dosage effect, whereby

a single 'faulty copy' of *TTN* is not sufficient to cause disease, but the additional deficiency in SRPK3 activity, affecting *TTN* transcriptional regulation and, in turn, normal full-length titin expression, tilts the scale toward pathology.

We have shown that, in vitro, SRPK3 phosphorylates at least one of the serine residues present in the transfected RBM20₅₁₇₋₆₆₄-V5 construct, corresponding to the RNA-recognition motif and

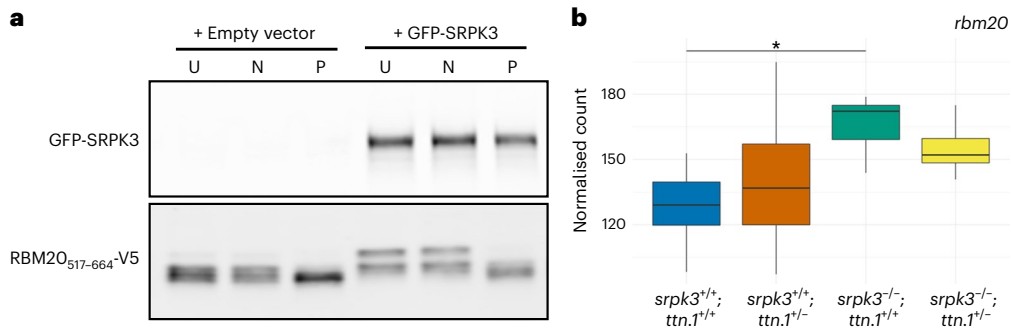

**Fig. 6 | SRPK3 phosphorylates RBM20 in vitro. a**, The RBM20$_{517-664}$-V5 reporter was transfected into 293T cells with or without GFP-SRPK3. GFP-SRPK3/RBM20$_{517-664}$-V5 co-expression resulted in RBM20$_{517-664}$-V5 hyperphosphorylation (lanes 4 and 5), as indicated by a mobility shift that was abolished by incubation with lambda phosphatase (P, lane 6). U indicates untreated samples; N indicates control samples incubated without phosphatase. In the absence of the SRPK3 construct, a less pronounced but still noticeable mobility shift can be observed (lanes 1 and 2), consistent with RBM20 phosphorylation by endogenous kinases such as SRPK1, CLK1 or AKT2. Assay was performed in quadruplicate. **b**, mRNA counts of the zebrafish RBM20 ortholog (BX649294.1 ENSDARG00000092881) are increased in *srpk3*-null zebrafish (*srpk3⁻/⁻*; *ttn.1⁺/⁺* and *srpk3⁻/⁻*; *ttn.1⁺/⁻*), likely as a feedback loop due to the *srpk3* deficiency. The box blots represent the first and third quartiles (25% and 75% percentile) with the center line at the median value. The whiskers extend from the hinge to the furthest value not beyond 1.5 times the interquartile range from the hinge. Differential expression was done using a two-sided Wald test with Benjamini–Hochberg adjustment for multiple testing[63]. For *srpk3⁻/⁻*; *ttn.1⁺/⁺* versus *srpk3⁺/⁺*; *ttn.1⁺/⁺*, *P* = 0.0379. *n* = 6 for each condition. Full-length blots are provided as source data.

RS-rich domains and including the RSRSP stretch. RSRSP phosphorylation is critical for RBM20 localization and activity[24,33]. RBM20 is a muscle-specific SR-splicing factor, primarily involved in I-band *TTN* alternative splicing, and known to regulate the ratio of N2BA:N2B cardiac isoforms[40]. Additional splicing targets include other sarcomeric genes (for example, *OBSCN* and *LDB3*), genes essential for calcium handling (for example, *CAMKD2* and *RYR2*) and even neuronal regulation (for example, *SEMA6D*)[41]. Mutations in *RBM20* have been associated with highly penetrant and severe dominant DCM in humans and other mammals[41,42]. Most frequently, these are gain-of-function missense changes in the highly conserved RSRPS region leading to cytoplasmic retention and aberrant ribonucleoprotein granules[24,43,44]. Conversely, loss-of-function (LoF) mutations outside the RSRSP stretch result in *RBM20* haploinsufficiency and aberrant splicing of target genes, such as *TTN*, but not mislocalization[45,46]. This is in line with population data showing *RBM20* to be highly LoF intolerant (pLI = 0.99)[18]. We manually interrogated the exome data of healthy *RBM20* LoF carriers and, interestingly, no cosegregating *TTN*tv were identified, just as seen with the *SRPK3* LoF hemizygotes.

It has been postulated that *RBM20*-DCM is more severe than *TTN*-DCM and thus cannot be solely explained by aberrant *TTN*-splicing regulation[47]. Notably, all but three of the *SRPK3/TTN* families did not present cardiac involvement. Speculatively, abnormal RBM20 phosphorylation by SRPK3 in the heart would be overcome by ubiquitously expressed kinases, as shown recently for cdc2-like kinases (CLKs) and protein kinase B (AKTs)[25] and supported by our in vitro phosphorylation assay. Although *RBM20* has not yet been associated with skeletal muscle disease, it has been shown to be DE across different skeletal muscles, where it regulates Z-band and M-band *TTN* splicing[48]. In addition, *TTN* RBM20-mediated splicing regulation is not only skeletal muscle-type specific but also affected by hormone levels[49] and circadian rhythm[50]. Overall, this highlights that *RBM20*-related pathology is complex and might be caused by the aberrant splicing of target genes through different tissue-specific genomic and nongenomic signaling pathways.

It is not clear whether the *SRPK3*-related myopathy is caused by the same pathomechanism in mice and humans. This type of discrepancy is not unique, and mouse models do not always recapitulate the human pathology, as seen, for example, in dystroglycanopathies[51,52]. Nevertheless, it is still possible that the identified *Ttn* variants might have an effect on the *Srpk3* KO mouse line. Interestingly, *Srpk3* overexpression in mice results in cardiomyopathy[16], not seen in the KO model. Like RBM20, SRPKs exhibit strong spatiotemporal expression profiles[13–16], and their subcellular localization is regulated by their own phosphorylation[53] and acetylation[54], suggesting that these kinases are involved in tightly controlled and fine-tuned pathways at different developmental stages and in response to external signals[17,55]. While their role in mRNA regulation is well studied[13], it has recently been shown that SRPKs are also involved in ubiquitin signaling[56].

We propose that the digenic inheritance of genes involved in post-translational processing and their direct or indirect targets[57] may be a model for conditions thus far thought to be monogenic[58]. Similar digenic inheritance models might also explain incomplete penetrance, such as recently described in spinocerebellar ataxia type 17 (ref. 59). In fact, *SRPK3* has previously been suggested as a causative gene in patients with X-linked spinocerebellar ataxia[60] and intellectual disability[34,61]. Notably, it has been proposed that abnormal SRPK-mediated phosphorylation of an E3 ubiquitin ligase might disrupt neurodevelopmental regulation[45,56]. In addition, haploinsufficiency of the splicing factor SRSF1, a well-known SRPK3 target[16], has been newly shown to cause a developmental disorder with intellectual disability[62]. Based on the findings presented here, it is conceivable that defective or absent phosphorylation activity of this kinase, in combination with a second deficient downstream target gene, could result in these, as well as other, disease phenotypes.

## Online content

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

Ana Töpf [1] ✉, Dan Cox [1], Irina T. Zaharieva[2], Valeria Di Leo[1,3], Jaakko Sarparanta [4,5], Per Harald Jonson [4,5], Ian M. Sealy [6,7], Andrei Smolnikov[8], Richard J. White [6,7], Anna Vihola[4,5,9], Marco Savarese[4,5], Munise Merteroglu [6,10], Neha Wali[11], Kristen M. Laricchia[12,13], Cristina Venturini[14], Bas Vroling[15], Sarah L. Stenton [12,16], Beryl B. Cummings[10], Elizabeth Harris [1,17], Chiara Marini-Bettolo[1], Jordi Diaz-Manera[1], Matt Henderson[18], Rita Barresi[19], Jennifer Duff[1], Eleina M. England[12], Jane Patrick[11], Sundos Al-Husayni[20], Valerie Biancalana[21], Alan H. Beggs [20], Istvan Bodi[22], Shobhana Bommireddipalli[23], Carsten G. Bönnemann [24], Anita Cairns[25], Mei-Ting Chiew[26], Kristl G. Claeys[27,28], Sandra T. Cooper [23], Mark R. Davis[26], Sandra Donkervoort[24], Corrie E. Erasmus[29], Mahmoud R. Fassad[30,31], Casie A. Genetti[20], Carla Grosmann[32], Heinz Jungbluth[33,34], Erik-Jan Kamsteeg [35], Xavière Lornage[21], Wolfgang N. Löscher[36], Edoardo Malfatti[37], Adnan Manzur[2], Pilar Martí [38,39], Tiziana E. Mongini[40], Nuria Muelas[38,39,41,42], Atsuko Nishikawa[43], Anne O'Donnell-Luria [12,16], Narumi Ogonuki[44], Gina L. O'Grady[45], Emily O'Heir [12], Stéphanie Paquay[46], Rahul Phadke[2], Beth A. Pletcher[47], Norma B. Romero[48], Meyke Schouten[35], Snehal Shah[49], Izelle Smuts[50], Yves Sznajer [51], Giorgio Tasca [1], Robert W. Taylor [30,31], Allysa Tuite [47], Peter Van den Bergh[46], Grace VanNoy[12], Nicol C. Voermans[52], Julia V. Wanschitz[36], Elizabeth Wraige[53], Kimihiko Yoshimura[54], Emily C. Oates[8], Osamu Nakagawa[55], Ichizo Nishino [43], Jocelyn Laporte [21], Juan J. Vilchez[38,39], Daniel G. MacArthur[12,56,57], Anna Sarkozy[2], Heather J. Cordell [58], Bjarne Udd[4,5,9], Elisabeth M. Busch-Nentwich[6,7], Francesco Muntoni[2,59] & Volker Straub [1] ✉

[1]John Walton Muscular Dystrophy Research Centre, Translational and Clinical Research Institute, Newcastle University and Newcastle Hospitals NHS Foundation Trust, Newcastle upon Tyne, UK. [2]Dubowitz Neuromuscular Centre, UCL Great Ormond Street Institute of Child Health & Great Ormond Street Hospital, London, UK. [3]Department of Life Sciences, University of Trieste, Trieste, Italy. [4]Folkhälsan Research Center, Helsinki, Finland. [5]Department of Medical and Clinical Genetics, Medicum, University of Helsinki, Helsinki, Finland. [6]School of Biological and Behavioural Sciences, Queen Mary University of London, London, UK. [7]Cambridge Institute of Therapeutic Immunology & Infectious Disease (CITIID), Department of Medicine, Jeffrey Cheah Biomedical Centre, University of Cambridge, Cambridge, UK. [8]School of Biotechnology and Biomolecular Sciences, University of New South Wales, Sydney, New South Wales, Australia. [9]Neuromuscular Research Centre, Tampere University and University Hospital, Tampere, Finland. [10]Laboratory of Angiogenesis and Cancer Metabolism, Department of Biology, University of Padua, Padua, Italy. [11]Wellcome Sanger Institute, Wellcome Genome Campus, Hinxton, UK. [12]Program in Medical and Population Genetics, Broad Institute of MIT and Harvard, Cambridge, MA, USA. [13]Analytic and Translational Genetics Unit, Massachusetts General Hospital, Boston, MA, USA. [14]Division of Infection and Immunity, University College London, London, UK. [15]Bio-Product, Nijmegen, The Netherlands. [16]Division of Genetics & Genomics, Department of Pediatrics, Boston Children's Hospital, Boston, MA, USA. [17]Northern Genetics Service, Institute of Genetics Medicine, Newcastle upon Tyne, UK. [18]Muscle Immunoanalysis Unit, Newcastle upon Tyne Hospitals NHS Foundation Trust, Newcastle upon Tyne, UK. [19]IRCCS San Camillo Hospital, Venice, Italy. [20]The Manton Center for Orphan Disease Research,

Division of Genetics and Genomics, Boston Children's Hospital, Harvard Medical School, Boston, MA, USA. [21]Institut de Génétique et de Biologie Moléculaire et Cellulaire (IGBMC), Inserm U1258, Cnrs UMR7104, Université de Strasbourg, Illkirch, France. [22]Department of Clinical Neuropathology, King's College Hospital NHS Foundation Trust, London, UK. [23]Kids Neuroscience Centre, the Children's Hospital at Westmead, the University of Sydney and the Children's Medical Research Institute, Westmead, New South Wales, Australia. [24]Neuromuscular and Neurogenetic Disorders of Childhood Section, National Institute of Neurological Disorders and Stroke, National Institutes of Health, Bethesda, MD, USA. [25]Neurosciences Department, Queensland Children's Hospital, Brisbane, Queensland, Australia. [26]Department of Diagnostic Genomics, PathWest Laboratory Medicine, Perth, Western Australia, Australia. [27]Department of Neurology, University Hospitals Leuven, Leuven, Belgium. [28]Laboratory for Muscle Diseases and Neuropathies, Department of Neurosciences, KU Leuven, Leuven, Belgium. [29]Department of Paediatric Neurology, Donders Institute for Brain, Cognition and Behavior, Radboud University Medical Centre, Amalia Children's Hospital, Nijmegen, The Netherlands. [30]Wellcome Centre for Mitochondrial Research, Translational and Clinical Research Institute, Faculty of Medical Sciences, Newcastle University, Newcastle upon Tyne, UK. [31]NHS Highly Specialised Service for Rare Mitochondrial Disorders, Newcastle upon Tyne Hospitals NHS Foundation Trust, Newcastle upon Tyne, UK. [32]Department of Neurology, Rady Children's Hospital University of California San Diego, San Diego, CA, USA. [33]Department of Paediatric Neurology, Neuromuscular Service, Evelina's Children Hospital, Guy's & St. Thomas' Hospital NHS Foundation Trust, London, UK. [34]Randall Centre for Cell and Molecular Biophysics, Muscle Signalling Section, Faculty of Life Sciences and Medicine (FoLSM), King's College London, London, UK. [35]Department of Human Genetics, Radboud University Medical Center, Nijmegen, The Netherlands. [36]Department of Neurology, Medical University Innsbruck, Innsbruck, Austria. [37]APHP, Neuromuscular Reference Center Nord-Est-Ile-de-France, Henri Mondor Hospital, Université Paris Est, U955, INSERM, Creteil, France. [38]Centro de Investigación Biomédica en Red de Enfermedades Raras (CIBERER), Madrid, Spain. [39]Neuromuscular Research Group, IIS La Fe, Valencia, Spain. [40]Department of Neurosciences Rita Levi Montalcini, Università degli Studi di Torino, Torino, Italy. [41]Department of Medicine, Universitat de Valencia, Valencia, Spain. [42]Neuromuscular Diseases Unit, Neurology Department, Hospital Universitari I Politècnic La Fe, Valencia, Spain. [43]Department of Neuromuscular Research, National Institute of Neuroscience, National Center of Neurology and Psychiatry, Tokyo, Japan. [44]RIKEN BioResource Research Center, Tsukuba, Japan. [45]Starship Children's Health, Auckland District Health Board, Auckland, New Zealand. [46]Cliniques Universitaires St-Luc, Centre de Référence Neuromusculaire, Université de Louvain, Brussels, Belgium. [47]Division of Clinical Genetics, Department of Pediatrics, Rutgers New Jersey Medical School, Newark, NJ, USA. [48]Neuromuscular Morphology Unit, Myology Institute, Sorbonne Université, Centre de Référence de Pathologie Neuromusculaire Nord/Est/Ile-de-France (APHP), GH Pitié-Salpêtrière, Paris, France. [49]Department of Neurology, Perth Children's Hospital, Nedlands, Western Australia, Australia. [50]Department of Paediatrics, Steve Biko Academic Hospital, University of Pretoria, Pretoria, South Africa. [51]Center for Human Genetic, Cliniques Universitaires Saint Luc, UCLouvain, Brussels, Belgium. [52]Department of Neurology, Donders Institute for Brain, Cognition and Behavior, Radboud University Medical Centre, Nijmegen, The Netherlands. [53]Evelina's Children Hospital, Guy's & St. Thomas' Hospital NHS Foundation Trust, London, UK. [54]Department Neurology, Nankoku Hospital, Kochi, Japan. [55]Department of Molecular Physiology, National Cerebral and Cardiovascular Center Research Institute, Osaka, Japan. [56]Centre for Population Genomics, Garvan Institute of Medical Research and UNSW, Sydney, New South Wales, Australia. [57]Centre for Population Genomics, Murdoch Children's Research Institute, Melbourne, Victoria, Australia. [58]Population Health Sciences Institute, Faculty of Medical Sciences, Newcastle University, Newcastle upon Tyne, UK. [59]NIHR Great Ormond Street Hospital Biomedical Research Centre, Great Ormond Street Institute of Child Health, UCL & Great Ormond Street Hospital Trust, London, UK. ✉e-mail: ana.topf@ncl.ac.uk; volker.straub@ncl.ac.uk

## Methods

No statistical methods were used to predetermine the sample size. The experiments were not randomized, and the investigators were not blinded to allocation during experiments and outcome assessment.

### Ethics statements

All clinical information and biological material used in this collaborative study were collected after obtaining written informed consent from the patients or their legal guardians. Each sequencing study was approved by the relevant health research authorities (Supplementary Note). Zebrafish were maintained in accordance with UK Home Office regulations, UK Animals (Scientific Procedures) Act 1986, under project licenses 70/7606 and P597E5E82. All animal work was reviewed by The Wellcome Trust Sanger Institute Ethical Review Committee.

### Genetic analysis

Genomic DNA from affected individuals was subjected to NGS and analyzed by applying standard filtering criteria (Supplementary Note). *SRPK3* and *TTN* variants were confirmed in the probands and assessed in available family members using Sanger sequencing. Deleteriousness of missense variants was predicted by Combined Annotation Dependent Depletion (CADD, v1.6) scores (https://cadd.gs.washington.edu/)[64]. NGS data were, when possible, analyzed for CNVs in the *TTN* gene and its triplicated region visually inspected. Variants in *SRPK3* were annotated based on the coding DNA reference sequence NM_014370 and transcript ENST00000370101. Variants in *TTN* were annotated based on NG 011618.3 or LRG 391 and inferred-complete variant-IC (NM 001267550.1 or ENST00000589042.5), usually referred to as the titin meta-transcript.

### Three-dimensional modeling of *SRPK3* variants

For the structure-based analysis of SRPK3 variants, a homology model was built using YASARA[65] (v15.4.10) with an SRPK1 structural template (5MYV, chain C).

### *SRPK3* and *TTN* gene expression and allelic balance analysis

RNA sequencing from muscle biopsies was performed at the Broad Institute Genomics Platform via the Tru-Seq Strand-Specific Large Insert RNA Sequencing protocol, at high coverage (50 M pairs). This included plating, poly-A selection and strand-specific cDNA synthesis, library preparation (450–550 bp insert size) and sequencing (101 bp paired reads). STAR-aligned BAMs were analyzed for gene expression and allelic balance. For gene expression analysis, the featureCounts[66] utility from the Subread package (v2.0.0) was used to count reads mapping to annotated genes across the human genome. Resultant counts were processed with edgeR[67] (v3.28.1), undergoing normalization with the calcNormFactors function followed by testing for differential gene expression between patient and control samples using the (two-sided) exactTest function[68]. Results for *SRPK3* and *TTN* were examined, and CPM values, normalized by gene length, were obtained with the cpm function and plotted.

Allelic balance for regions of *TTN* with evidence of heterozygosity was assessed using the AllelicImbalance[69] (v1.24.0) software package, which counts reads at every heterozygous position and applies a chi-square test to determine the statistical significance of any deviation from the expected ratio of reads at each position.

### Muscle biopsy and MRI analysis

Muscle biopsies were analyzed following standard histological techniques for light and electron microscopy as part of the diagnostic workup of patients in accredited pathology laboratories. Muscle MRIs were obtained on standard diagnostic scanners using axial T1-weighted scans.

### Titin immunoanalysis

Western blotting of titin C-terminal fragments was carried out as previously described[20], using two different antibodies raised against the C-terminal M10 Ig domain of titin, rabbit polyclonal M10-1 (ref. 9; 1:300) and mouse monoclonal 11-4-3 (ref. 20; 1:150). For high molecular weight titin western blotting, antibody Z1Z2 (1:1,500, N-terminal, TTN-1; Myomedix) and F146.9B9 (1:1,000, distal I-band titin; Enzo Life Sciences) were used. Snap-frozen muscle biopsies were homogenized in a sample buffer containing 8 M urea, 2 M thiourea, 10% SDS, 0.05 M Tris base and 10% glycerol supplemented with 10% β-mercaptoethanol and heated at 60 °C for 15 min. The soluble fraction was recovered after centrifugation. Equal amounts of muscle protein were loaded into vertical 1% agarose gels, and the run was performed at +8 °C, using 12.5 mA per gel for 6–7 h. The proteins were detected in-gel using SimplyBlue SafeStain (Invitrogen) or blotted to PVDF membranes using CAPS buffer in a Trans-Blot Turbo System with 40 V for 5 h.

### Statistical analysis

Three cohorts of patients with genetically confirmed forms of limb-girdle muscular dystrophy, namely LGMD-R1 ($n = 170$), LGMD-R2 ($n = 94$) and LGMD-R12 ($n = 56$) caused by recessive mutations in the *CAPN3*, *DYSF* and *ANO5* genes, respectively, were identified through the MYO−SEQ Project[70]. The number of *TTN*tvs (that is, stop gain, splice sites and frameshift) was counted in each disease control population. A Fisher's test (two-sided, with no adjustment for multiple comparisons), following the proposal of Agresti and Coull[71] to add two successes and two failures to each data set was calculated.

Similar statistical analysis was implemented for the comparison between the number of *TTN*tvs found in affected and unaffected males carrying a truncating *SRPK3* variant (referred to as '*SRPK3*-null males'; Supplementary Note). For the statistical modeling, the MLINK (v5.10) program from the LINKAGE[72] package and PSEUDOMARKER[73,74] (v2.0) were used. For details, see Supplementary Note.

### Zebrafish husbandry and genotyping

Zebrafish were maintained in accordance with UK Home Office regulations, UK Animals (Scientific Procedures) Act 1986, under project licenses 70/7606 and P597E5E82. All animal work was reviewed by The Wellcome Trust Sanger Institute Ethical Review Committee. Zebrafish were maintained at 23.5 °C on a 14 h light/10 h dark cycle. The mutant lines *srpk3^sa18907^* and *ttn.1^sa5562^* were generated by the Zebrafish Mutation Project[75]. The allele *srpk3^sa18907^* is an essential splice site mutation affecting the donor site of exon 15, and *ttn.1^sa5562^* is a premature stop codon (zebrafish assembly GRCz11 chr9:42861631T>G) in exon 19 of 214, corresponding to amino acid p.Leu1570. Genotyping was carried out as previously described[76]. Filamentous actin and α-actinin were stained in genotyped larvae at 5 dpf using Alexa Fluor 488 phalloidin (Invitrogen, A12379; 1:80) and mouse monoclonal anti-α-actinin (Sigma, A7811; 1:200).

### Zebrafish transcriptomics

In total, 5-dpf larvae were collected as six pools of three embryos per genotype to minimize any differences due to biological variance. A previously described protocol for single embryo RNA extraction[77] was optimized for use with pools of zebrafish larvae. Samples were lysed in 110 μl RLT buffer (RNeasy kit, Qiagen) containing 1.1 μl of 14.3 M β-mercaptoethanol (Sigma). The lysate was allowed to bind to 450 μl of Agencourt AMPure XP beads (Beckman Coulter) for 15 min. The tubes were left on a magnet (Invitrogen) until the solutions cleared, and the supernatant was then removed without disturbing the beads. While still on the magnet, the beads were washed three times with 70% ethanol and allowed to dry for 20 min. Total nucleic acid was eluted from the beads following the manufacturer's instructions and treated with DNase I (New England Biolabs, M0303L). RNA was quantified using a NanoDrop (Thermo Fisher Scientific NanoDrop One Microvolume

UV–Vis Spectrophotometer), RNA integrity numbers were checked using a Bioanalyzer (2100 Bioanalyzer System) and sequencing libraries were made using the NEBNext Ultra II DNA Library Prep Kit for Illumina.

Libraries were pooled and sequenced on one lane of NovaSeq 6000 SP PE150 (between 14 million and 29 million reads per sample). RNA-seq data have been deposited in the ArrayExpress database at EMBL-EBI (www.ebi.ac.uk/arrayexpress) under accession E-MTAB-12934. Sequencing data were assessed using FastQC (v0.11.9; https://www.bioinformatics.babraham.ac.uk/projects/fastqc/) and aligned to the GRCz11 reference genome using STAR[78] (v2.7.3a). Read counts per gene were generated by STAR and used as input for differential expression analysis using the R package DESeq2 (v1.28.1)[79]. The following model was used for DESeq2: ~Genotype, modeling counts as a function of the genotype. Genes with an adjusted $P$ value of <0.05 using a two-sided Wald test with Benjamini–Hochberg adjustment[63] for multiple testing were considered to be DE. Gene sets were analyzed using topGO[80] (v2.38.1), and the resulting GO enrichments were visualized using the ClueGO (v2.5.9) plugin for Cytoscape[81] (v3.9.1). For analysis of gene expression changes and visualization of data, R (v4.2.0; R Core Team; https://www.r-project.org/) was used, together with the tidyverse[82] package (v1.3.1). Differential alternative splicing events were analyzed using rMATS[83] (v4.1.2).

### Myofiber immunofluorescence

Immunofluorescence staining of zebrafish skeletal myofibers was performed by adapting a protocol previously described[84] (Supplementary Note). Myofibers were incubated in primary antibody overnight at 4 °C, washed in 1x Phosphate-Buffered Saline with 0.1% Tween 20 (PBST), then incubated with goat anti-mouse Alexa Fluor 488 secondary antibody (Thermo Fisher Scientific, A11001; 1:500) and goat anti-rabbit Alexa Fluor 594 secondary antibodies (Thermo Fisher Scientific, A11037; 1:250) for 1 h at room temperature. Further washing in PBST was performed before mounting with Vectashield Mounting Medium (Vector Laboratories). Primary antibodies used were rabbit polyclonal anti-sarcomeric α-actinin (Cell Signaling Technologies, 3134; 1:25) and mouse monoclonal anti-titin (Merck, T9030; 1:200). Digital images were captured with Newcastle University Bioimaging Unit's Nikon A1R point scanning confocal microscope (Nikon) using Nikon Elements AR (v5.21.03).

### Transmission electron microscopy (TEM)

Genotyped zebrafish embryos (5 dpf) were fixed with glutaraldehyde (2%) fixative in 0.1 M cacodylate buffer, pH 7.4 (2BScientific, 30450003-1) at 4 °C. Processing for TEM was performed by Newcastle University Electron Microscopy Research Services. Ultrathin sections were stained with heavy metal salts (uranyl acetate and lead citrate). Sections were imaged on a Hitachi HT7800 120 kV TEM using an EMSIS CMOS Xarosa high-resolution camera (Hitachi) and Radius software (v2.0).

### Mouse whole-genome sequencing (WGS) analysis

DNA from *Srpk3*-null (KO)[16] and WT (129s6/SvEvTAC background) mice were subjected to WGS at deCODE Genetics. Fastq files were mapped using the BBmap suite (v38.69) against the mouse *Ttn* gene sequence (USCS, GRCm38/mm10). Variants (with a read frequency of >20%) were called using Varscan 2 (v2.3.7) for each sample, and BCFTools (v1.9) was used to merge all samples by group (WT and KO). The output variants present in the WT and KO groups were annotated using the Variant Effect Predictor tool (Ensembl, https://www.ensembl.org/info/docs/tools/vep/index.html, https://www.ensembl.org/Tools/VEP). Variants were compared to the reference 129s6/SvEvTAC background.

### Constructs

To create the RBM20$_{517-664}$-V5 reporter, the cDNA fragment coding amino acids 517–664 of mouse RBM20 was cloned into pcDNA3.1D/V5-His-TOPO in frame with C-terminal V5 and His6 tags using the

pcDNA3.1 Directional TOPO Expression Kit (Thermo Fisher Scientific, K490001). The pcDNA5-FRT/TO-GFP-SRPK3 construct (DU 25699), expressing N-terminally GFP human SRPK3, was obtained from the Medical Research Council Protein Phosphorylation and Ubiquitylation Unit Reagents and Services (University of Dundee, Scotland).

### Phosphorylation assay

The 293T cells were transfected with the RBM20$_{517-664}$-V5 reporter construct together with either GFP-SRPK3 or an empty vector (pcDNA5/TO), collected after 2 d of expression and frozen at −80 °C. The cells were lysed for 15 min on ice in 1× NEBuffer Pack for Protein Metallophosphatases (50 mM HEPES (pH 7.5), 100 mM NaCl, 2 mM DTT, 0.01% Brij 35; New England Biolabs) supplemented with 1 mM MnCl$_2$, 1% Triton X-100, 1× Halt Protease Inhibitor Cocktail (Thermo Fisher Scientific), 4 mM β-glycerophosphate, 10 mM sodium pyrophosphate, 4 mM sodium orthovanadate and 2 mM sodium fluoride. Insoluble material was pelleted at 4 °C for 15 min at 15,700$g$. The supernatants were divided into three reactions (U, untreated; N, no phosphatase; P, phosphatase) and combined 1:1 with dephosphorylation buffer (1× NEBuffer pack for protein metallophosphatases, 1 mM MnCl$_2$) alone (U, N) or with NEB lambda protein phosphatase (P; final concentration 6,667 units ml$^{-1}$). The U reactions were immediately mixed with 2× SDS sample buffer and heated for 5 min at 95 °C, whereas the N and P reactions were incubated for 30 min at 30 °C before SDS sample preparation. The samples were run in standard SDS–PAGE in 4–20% TGX minigels (Bio-Rad), transferred on nitrocellulose membranes and stained with antibodies against the V5 tag (Thermo Fisher Scientific, R960-25, SV5-Pk1; 1:5,000) and GFP (Thermo Fisher Scientific, A-11122; 1:5,000).

### Reporting summary

Further information on research design is available in the Nature Portfolio Reporting Summary linked to this article.

## Data availability

Due to privacy, ethical and legal issues de-identified patient genomic, transcriptomic and phenotypic data that supports the findings of this study can only be available from the corresponding author upon reasonable request. Zebrafish RNA-seq data can be accessed in the ArrayExpress database at EMBL-EBI (www.ebi.ac.uk/arrayexpress) under accession E-MTAB-12934. Mouse WGS data and human RNA-seq data can be accessed in the Sequence Read Archive under accession (PRJNA1027609 and PRJNA1027754, respectively). Control frequencies and variant information were extracted from gnomAD (v2.1.1; https://gnomad.broadinstitute.org). *TTN* variant information was obtained from the Leiden Open Variation Database (https://databases.lovd.nl/shared/genes/TTN). Source data are provided with this paper.

## Code availability

All software used to analyze the study data are listed in the Methods and in the Nature Research Reporting Summary and are publicly available.

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

## Acknowledgements

We acknowledge H. Luque, L. Phillips, J. Casement, O. Magnuson, D. Nguyen and Y. Hu for technical support; R. García-Tercero and C. Díaz for sample collection; E. Zorio, M.E. Leach, D. Bharucha-Goebel, J. Dastgir and C. Konersman for clinical expertise and M. Gautel for helpful advice. We also thank CureCMD for their help in patient recruitment and the patients for donating their samples.

The research leading to these results has received funding from the European Community's Seventh Framework Program (FP7/2007-2013; 2012-305121) 'Integrated European—omics research project for diagnosis and therapy in rare neuromuscular and neurodegenerative diseases (NEUROMICS)' (to A. Töpf, V.S., I.T.Z. and F.M.); the European Union's Horizon 2020 research and innovation program (Solve-RD project; 779257 to A. Töpf); Muscular Dystrophy UK and Muscular Dystrophy Association US (mda577346 to F.M.); Päulon Säätiö (to M. Savarese); Academy of Finland, Sigrid Juselius Foundation (to B.U.); core funding to the Sanger Institute by the Wellcome Trust (098051 and 206194 to E.M.B.-N., J.P. and N.W.); EURO-NMD and Fundación Gemio (to J.J.V., N.M. and P.M.); Intramural Research Grant (2-5, 29-4) for Neurological and Psychiatric Disorders of NCNP and AMED (JP20ek0109490h0001 to I.N.); Inserm, CNRS, University of Strasbourg, Labex INRT (ANR-10-LABX-0030 and ANR-10-IDEX-0002-02), France Génomique (ANR-10-INBS-09) and Fondation Maladies Rares for the 'Myocapture' sequencing project, AFM-Téléthon (22734), the European Joint program (EJPRD2019-126 IDOLS-G and ANR-19-RAR4-0002 to J.L., X.L. and V.B.); Intramural funds from the NIH National Institute of Neurological Disorders and Stroke (to C.G.B.); the Dutch Princess Beatrix Muscle Fund and the Dutch Spieren voor Spieren Muscle fund (to C.E.E.); PI16/00316 supported by the Instituto de Salud Carlos III (ISCIII), Madrid and the Generalitat Valenciana (grant PROMETEO/2019/075 to N.M.); Australian NHMRC Neil Hamilton Fairley Early Career Research Fellowship (GNT1090428 to E.C.O.); Starship Foundation A+7340 (to G.L.O.); Early Career Award from the Thrasher Research Fund (to S.S.); U54 HD090255 from the NIH Eunice Kennedy Shriver National Institute of Child Health and Human Development (to A.H.B.); Wellcome Center for Mitochondrial Research (203105/Z/16/Z), the Mitochondrial Disease Patient Cohort (UK; G0800674), the Medical Research Council International Center for Genomic Medicine in Neuromuscular Disease (MR/S005021/1), the Medical Research Council (MR/W019027/1), the Lily Foundation, Mito Foundation, the Pathological Society, the UK NIHR Biomedical Research Center for Ageing and Age-related Disease award to the Newcastle upon Tyne Foundation Hospitals NHS Trust and the UK NHS Highly Specialized Service for Rare Mitochondrial Disorders of Adults and Children (to R.W.T.). MYO–SEQ was funded by Sanofi Genzyme, Ultragenyx, LGMD2I Research Fund, Samantha J Brazzo Foundation, LGMD2D Foundation, Kurt+Peter Foundation, Muscular Dystrophy UK and Coalition to Cure Calpain 3. Sequencing and analysis for relevant families (Supplementary Note) were provided by the Broad Institute of MIT and Harvard Center for Mendelian Genomics (Broad CMG) and were funded by the National Human Genome Research Institute, the National Eye Institute and the National Heart, Lung and Blood Institute under grant UM1 HG008900 and the National Human Genome Research Institute under grants U01HG0011755 and R01 HG009141. The content is solely the responsibility of the authors and does not necessarily represent the official views of the National Institutes of Health. DNA samples for NeurOmics and MYO–SEQ were provided by the John Walton Muscular Dystrophy Research Center Biobank. This facility is supported by the NIHR Newcastle Biomedical Research Center. Newcastle University's Electron Microscopy Research Services and equipment Hitachi HT7800 120 kV TEM microscope are funded by BBSRC grant reference BB/R013942/1.

## Author contributions

A. Töpf, B.U., E.M.B.-N., F.M., H.J.C., J.L., I.N. and V.S. designed the study. Data were collected by A. Töpf, D.C., A.H.B., A.C., A.M., A.N., A.O'D.-L., A. Sarkozy, A. Tuite, B.A.P., B.V., C.E.E., C.A.G., C.G., C.G.B., C.M.-B., C.V., D.G.M., E.C.O., E.H., E.-J.K., E.M., E.M.E., E.O'H., E.W., G.L.O'G., G.T., G.V., H.J., H.J.C., I.B., I.N., I.S., I.T.Z., J.D., J.D.-M., J.J.V., J.L., J.P., J.S., J.V.W., K.G.C., K.Y., M.H., M.M., M.R.D., M.R.F., M. Savarese, M. Schouten, M.-T.C., N.C.V., N.M., N.O., N.B.R., N.W., O.N., P.M., P.V.d.B., R.B., R.P., R.W.T., S.A.-H., S.T.C., S.D., S.L.S., S.P., S.S., T.E.M., V.B., V.D.L., W.N.L., X.L. and Y.S. Formal analysis was carried out by A. Töpf, A. Smolnikov, B.B.C., B.V., D.C., E.M.B.-N., H.J.C., I.M.S., J.P., K.M.L., M.M., N.W., P.H.J., R.J.W. and S.L.S. Visualization of data was done by A. Töpf, A. Smolnikov, A.V., D.C., I.M.S., I.T.Z., J.D.M., J.S., J.V.W., M.M., P.H.J., R.J.W. and S.B. A. Töpf wrote the original draft. All authors reviewed and edited the draft and approved the final version of the manuscript.

## Competing interests

The authors declare no competing interests.

## Additional information

**Extended data** is available for this paper at

**Supplementary information** The online version
contains supplementary material available at

**Correspondence and requests for materials** should be addressed to
Ana Töpf or Volker Straub.

**Peer review information** *Nature Genetics* thanks James Dowling and the
other, anonymous, reviewer(s) for their contribution to the peer review of
this work. This article has been peer-reviewed as part of Springer Nature's
Guided Open Access initiative. Peer reviewer reports are available.

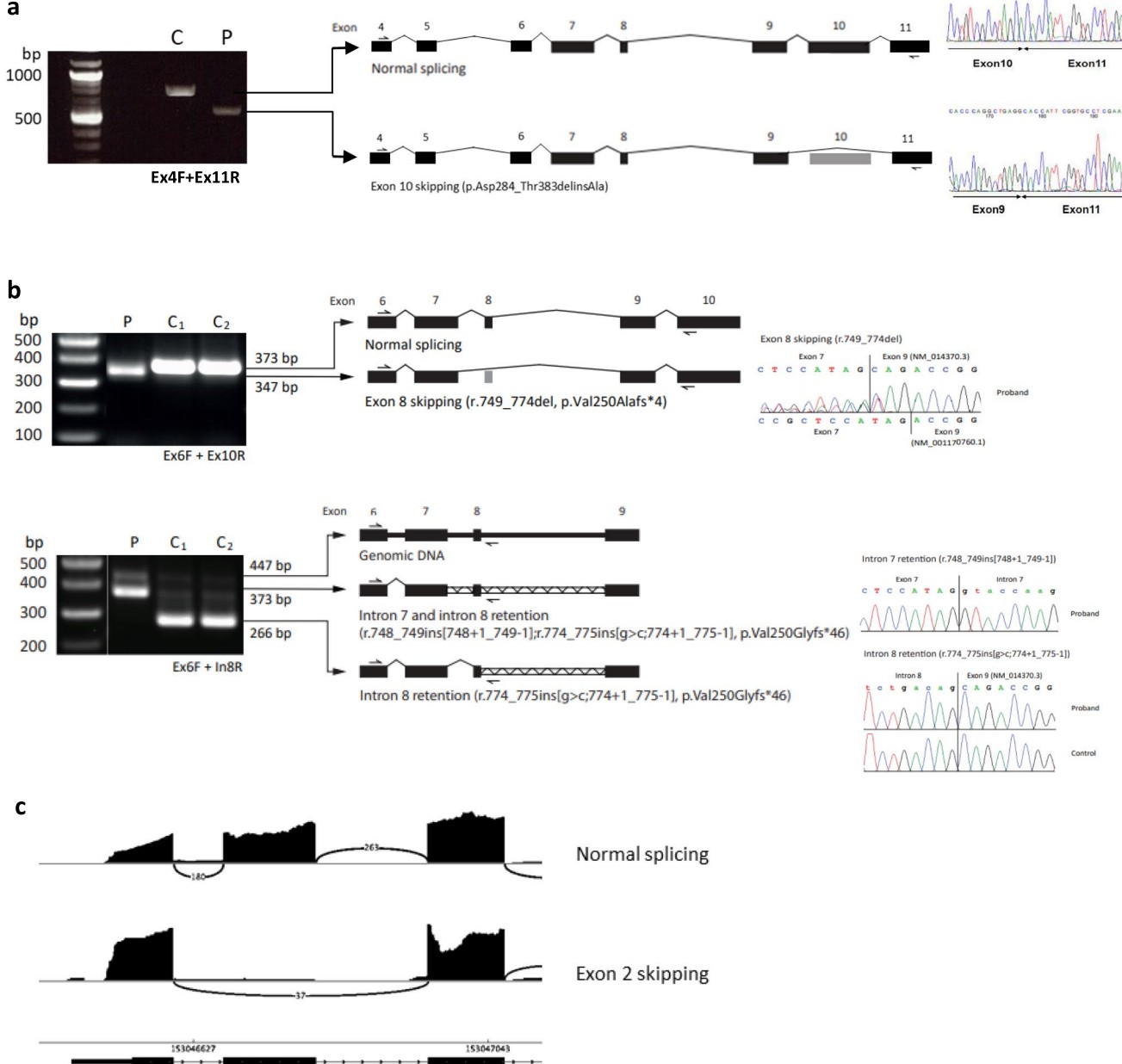

**Extended Data Fig. 1 | RNA analysis of *SRPK3* truncating variants. a**, RT-PCR of muscle-derived cDNA from patient CII:2 carrying a donor splice site variant (c.1144+1G>A). PCR amplification with Ex4F/Ex11R primers showed that the variant leads to the exclusion of exon 10 and results in a frameshift change (p.Asp284_Thr383delinsAla). **b**, RT-PCR of muscle-derived cDNA from patient SII:1 carrying an extended splice site variant (c.774+5G>C); top: using primers in exons flanking the variant (Ex6F 5′-CGTGAAGAGCATCGTGAGG and Ex10R 5′- GCCCCCGTCTAGTCTCAAG) a single band was detected in the patient (P), corresponding to abnormal exon 8 skipping (r.749_774del, p.Val250Alafs*4); bottom: using the same forward primer in exon 6 and a reverse primer in intron 8 (In8R 5′- GACGGCCCGGTACTGCCGAGTCTG), two and three bands were detected

in the proband and control samples, respectively. The larger bands correspond to gDNA; the band at 373 bp corresponds to abnormal retention of both intron 7 and intron 8 in the proband (r.748_749ins[748+1_749-1]; r.774_775ins[g>c;774+1_775-1], p.Val250Glyfs*46). The lower band corresponds to a natural missplicing event leading to retention of intron 8 observed both in the patient and controls (r.774_775ins[g>c;774+1_775-1], p.Val250Glyfs*46). **c**, Sashimi plot for RNA sequencing data from muscle tissue from patient LII:1 carrying a donor splice variant (c.190+2T>C). The plot shows skipping of exon 2, leading to an out-of-frame mRNA. Any *SPRK3* transcripts escaping nonsense-mediated decay will encode a truncated protein lacking the kinase domain. RT-PCR amplification was performed at least in duplicate.

## *SRPK3* (ENST00000370101.3)

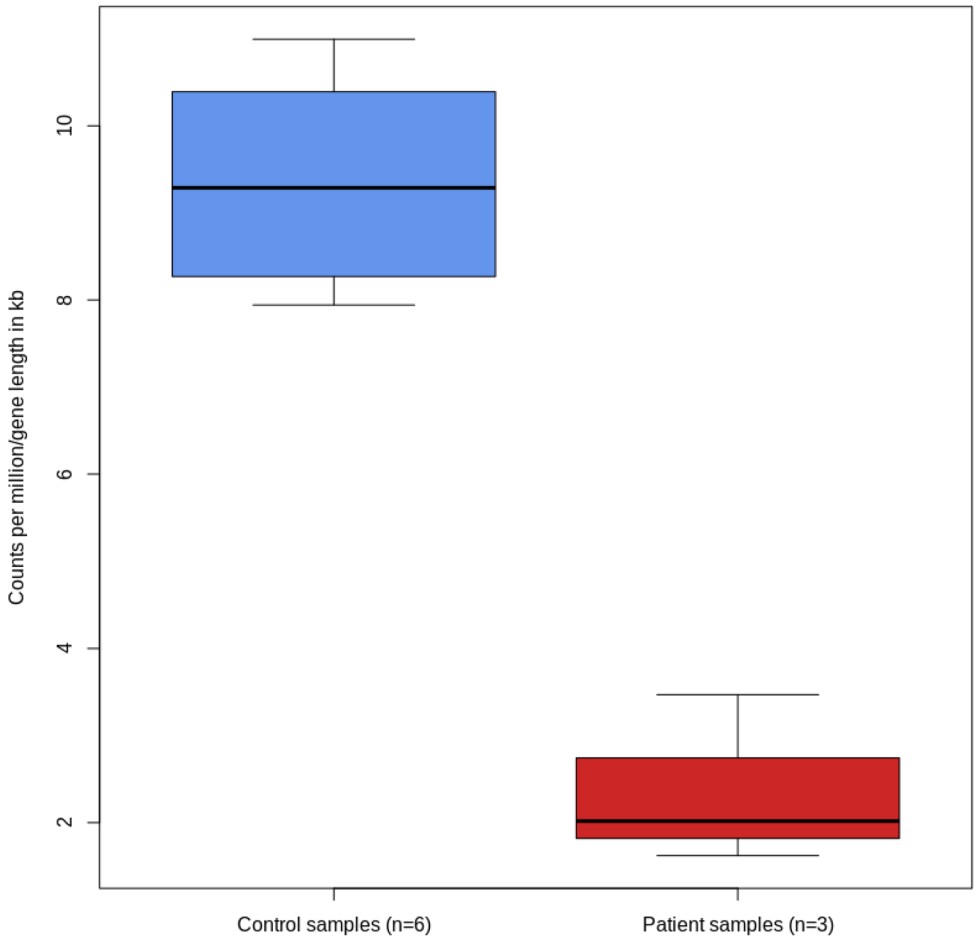

**Extended Data Fig. 2 | *SRPK3* mRNA expression levels in patients with *SRPK3* truncating variants.** RNA sequencing data from patients carrying *SRPK3* truncating variants (LII:1, DII:1 and YII:3) were analyzed for *SRPK3* expression between patient ($n = 3$) and control ($n = 6$) samples. Counts per million (CPM) values were normalized by gene length. The boxes represent the first and third quartiles (25% and 75% percentile) with the center line at the median value. The whiskers extend from the hinge to the furthest value not beyond 1.5 times the interquartile range from the hinge. Differential expression was performed in edgeR using a two-sided exact test, with no adjustment for multiple comparisons. Uncorrected *P*-value = $7.106 \times 10^{-11}$.

a

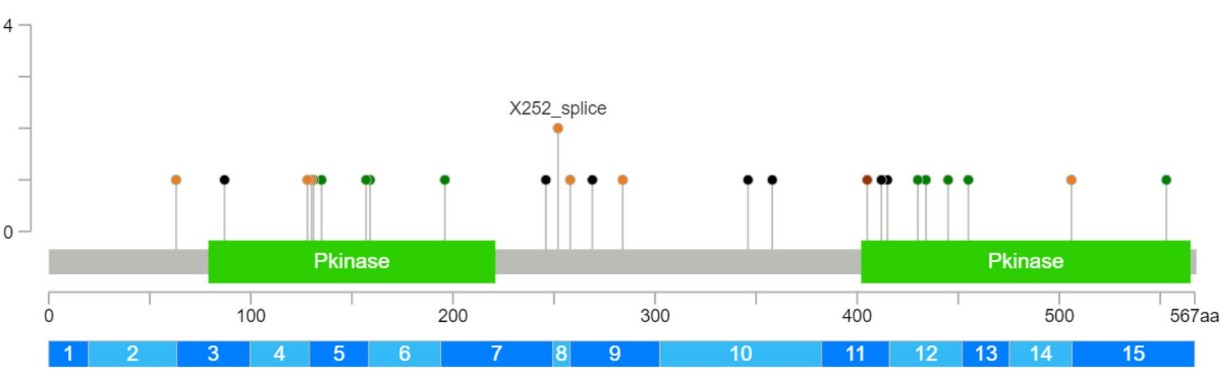

b

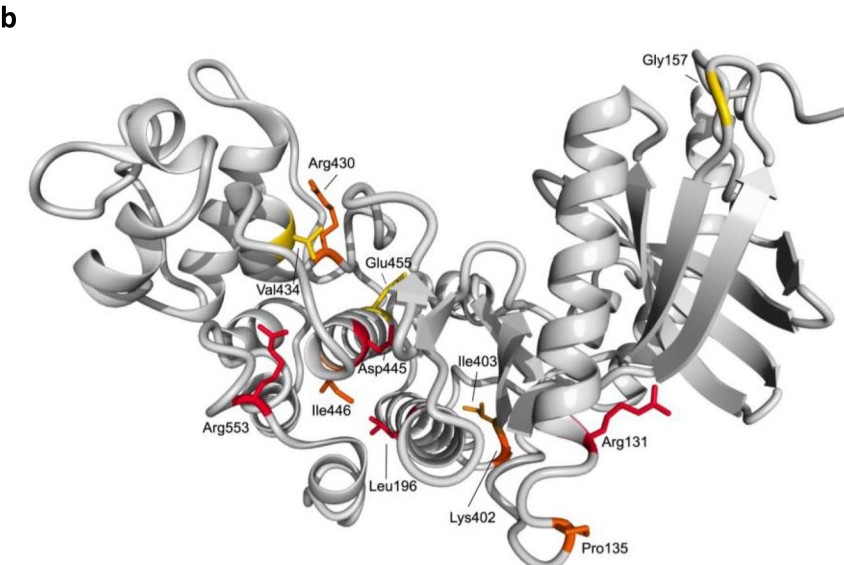

**Extended Data Fig. 3 | Localization and 3D modeling of *SRPK3* variants.**
**a**, Localization of *SRPK3* variants. Kinase domains are indicated in green.
Missense changes are shown in green, truncating variants (nonsense, splice
sites and frameshift) in black and the in-frame variant in red. Two splice site
variants were identified at position c.749-2 (shown as X252_splice). **b**, For the
structure-based analysis of *SRPK3* variants, a homology model was built using
YASARA (v15.4.10) with a SRPK1 structural template (5MYV, chain C). Amino
acid position for changes p.Arg131Pro, p.Pro135His, p.Gly157Arg, p.Leu196Pro,
p.Arg430Gln, p.Val434Glu, p.Asp445Asn, p.Glu455Lys, p.Arg553Trp and the
p.Lys405_Ile406del are indicated. p.Arg131Pro + p.Pro135His: Pro135 is located in
a loop where it bends the loop in a way to allow for stabilizing interactions, where
a change to His will lead to disruption of local loop orientation. Arg131 is located
at the C-terminal of the alpha-helix, close to Pro135. In an additive fashion, the
introduction of Arg131Pro will probably destabilize local structure even further.
p.Gly157Arg: will force changes in backbone orientation for residues in the loop
and surrounding sheet structures, leading to local stability issues by forcing
surrounding residues to adopt orientations that impair favorable interactions.
p.Leu196Pro: will lead to a disruption of the helical structure and decrease
protein stability. p.Arg430Gln: Arg430 plays a stabilizing role by interacting

with the negatively charged residues Asp474 and Asp423. Loss of this positive
charge will lead to destabilizing effects. p.Val434Glu: introduces a negatively
charged residue that will likely disrupt the wild-type charged interaction
network of Arg430, Asp474 and Asp423. p.Asp445Asn: loss of negative charge,
loss of hydrogen bond with backbone Thr211, leading to a destabilizing effect.
p.Glu455Lys: change of a highly conserved negative charge into positive charge.
Forms a hydrogen bond with Tyr429, which is lost when mutated to Lys. Probably
destabilizing, although the role of the negative charge is not clear. p.Arg553Trp:
Arg533 forms a salt bridge with Glu433, stabilizing local structure. Losing
that salt bridge will destabilize. Additionally, exchanging a large, positively
charged residue for a bulky very hydrophobic residue will lead to additional
destabilization. p.Lys405_Ile406del (also annotated as p.Lys402Ile403del):
results in a deletion of two residues (Lys-Ile) in a short repeat sequence (Lys-Ile-
Lys-Ile-Lys-Ile). Protein structure modeling suggests that the deleted residues
are Lys402 and Ile403. The modeled structure shows Asp401 reoriented into
a position originally occupied by Ile403. This will destabilize local structure.
Additionally, it results in the loss of the salt bridge between Asp401 and Arg193,
which also contributes to a destabilizing effect.

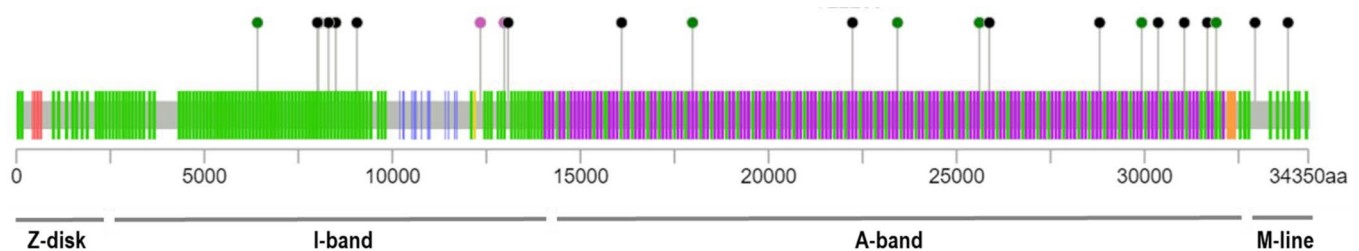

**Extended Data Fig. 4 | Distribution of *TTN* variants identified in the *SRPK3*/*TTN* cohort.** *TTN* truncating variants (nonsense, splice sites and frameshift) are shown in black and missense variants in green. No missense variants in *TTN* exons 344 or 364, associated with HMERF and tibial muscular dystrophy, respectively, were found. *TTN* variants were located mainly in the A-band and I-band; however, no clustering was observed. Two frameshift variants occurred in meta transcript only exons (in pink).

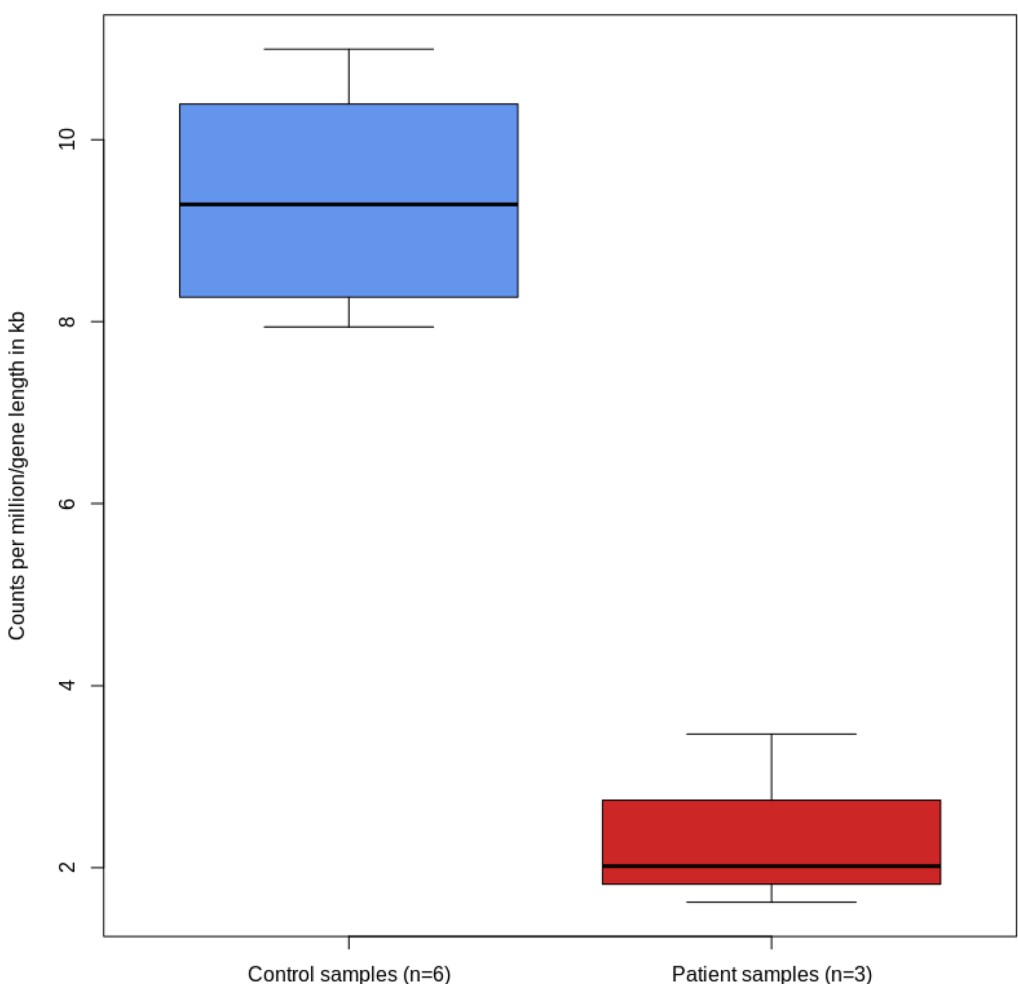

*TTN* (ENST00000589042.5)

**Extended Data Fig. 5 | *TTN* mRNA expression levels in *SRPK3/TTN* patients.** RNA sequencing data from *SRPK3/TTN* myopathy patients (LII:1, DII:1 and YII:3) were analyzed for *TTN* expression between patient (*n* = 3) and control (*n* = 6) samples. Counts per million (CPM) values were normalized by gene length. The boxes represent the first and third quartiles (25% and 75% percentile) with the center line at the median value. The whiskers extend from the hinge to the furthest value not beyond 1.5 times the interquartile range from the hinge. Differential expression was performed in edgeR using a two-sided exact test, with no adjustment for multiple comparisons. Uncorrected *P*-value = 0.0008015. This was likely not due to nonsense-mediated mRNA decay, as there was no evidence of allele-specific expression at any of the heterozygous *TTN* loci examined.

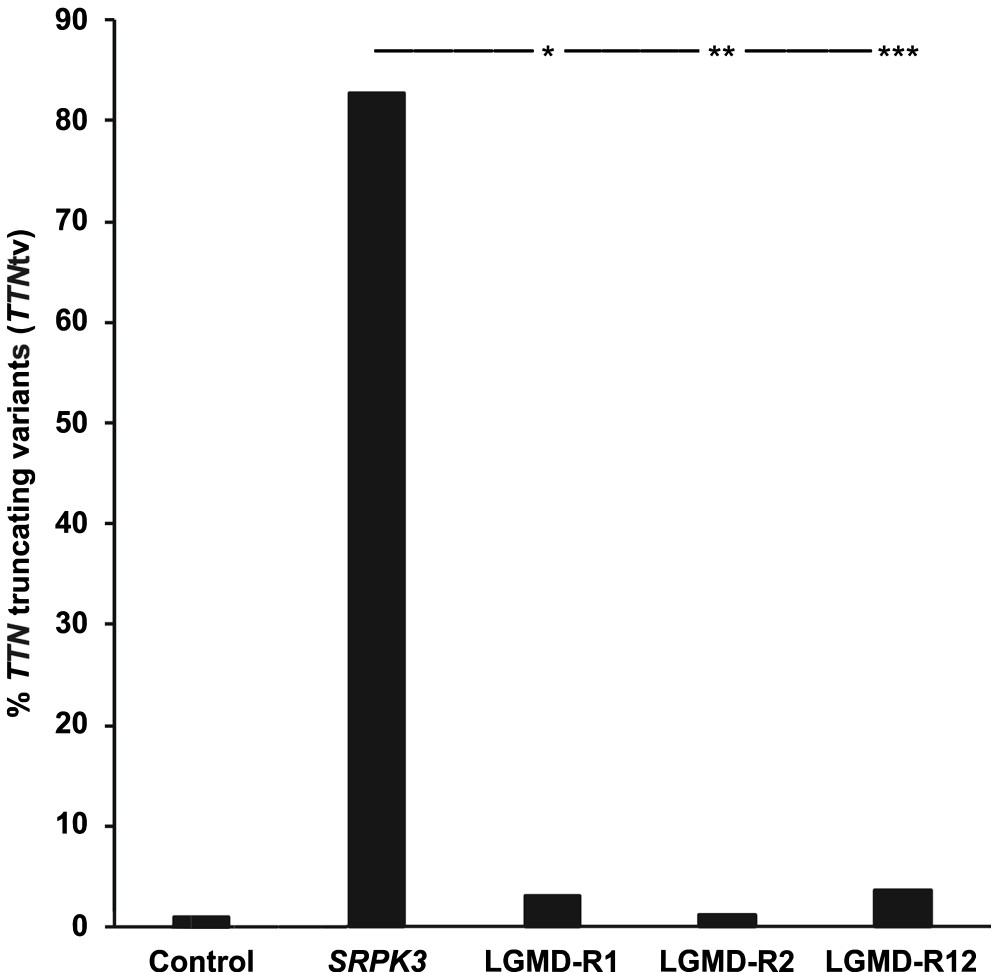

**Extended Data Fig. 6 | *TTN*tv in other muscle disease populations.** Comparison between the number of *TTN*tv (stop gain, splice sites and frameshift variants) in the *SRPK3/TTN* myopathy families ($n = 25$) and three cohorts of patients with genetically confirmed forms of limb girdle muscular dystrophy: LGMD-R1 ($n = 170$), LGMD-R2 ($n = 94$) and LGMD-R12 ($n = 56$). A Fisher's test (two-sided, with no adjustment for multiple comparisons), following the proposal of Agresti and Coull to add two successes and two failures to each data set was calculated. *P*-values: (*) $= 6.13 \times 10^{-19}$, (**) $= 9.18 \times 10^{-17}$ and (***) $= 4.89 \times 10^{-12}$.

**a**

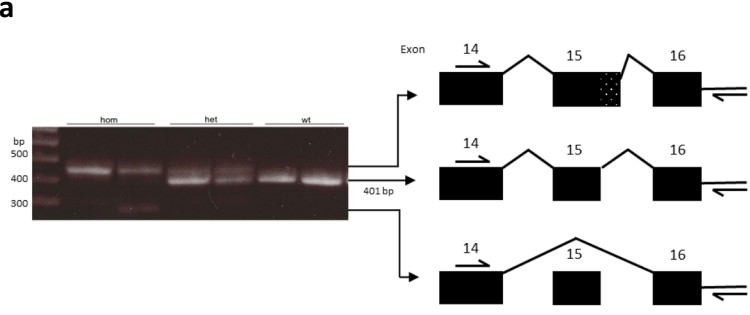

**b**

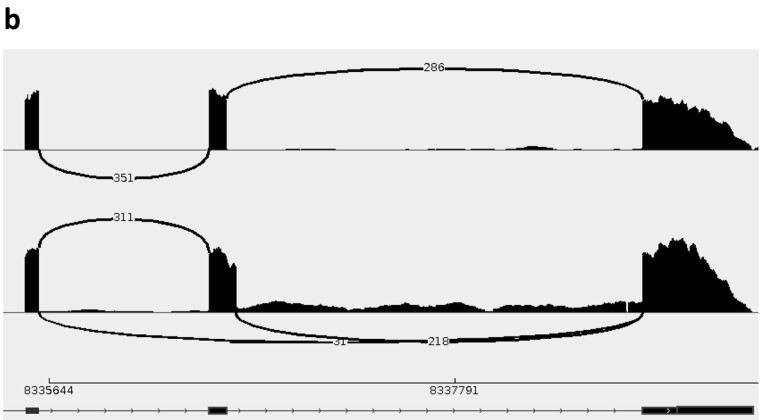

**Extended Data Fig. 7 | Effect of the zebrafish *srpk3* sa18907 mutation on mRNA. a**, The mutation lies between exon 15 and 16 in the zebrafish genomic sequence. RT-PCR of muscle-derived cDNA from zebrafish (wild-type, heterozygous and homozygous for the sa18907 mutation) showed three different sized products (primers: Fwd 5′- CTGCTGACATATGGAGCACTG and Rev 5′-GGATACTAAATGTCCCGTAGGTTG). Wild-type samples showed the expected 401-bp product (middle band). The mutation results in aberrant splicing of *srpk3* with partial retention of intron 15 (top band) seen both in the homozygous and heterozygous state, or loss of exon 15 (lower band) only seen in the homozygous mutants. Representative of two experiments. **b**, Sashimi plot for RNA sequencing data from *srpk3* mutant and wild-type zebrafish, also showing skipping of exon 15, as well as several forms of partial retention of intron 15.

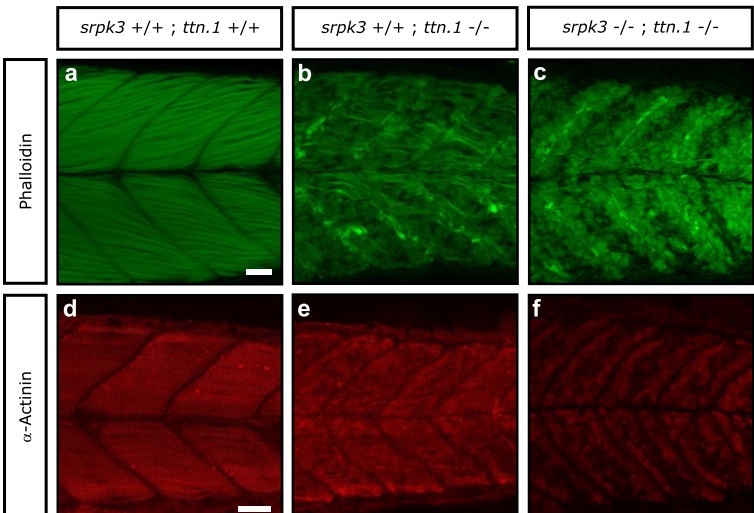

**Extended Data Fig. 8 | *ttn.1*-null zebrafish shows a severe skeletal muscle phenotype. a–f**, Lateral view of Alexa Fluor phalloidin filamentous actin (green) and α-actinin Z-band marker (red) staining in skeletal fast muscle fibers in wild-type (**a**,**d**), *srpk3*$^{+/+}$;*ttn.1*$^{-/-}$ (**b**,**e**) and *srpk3*$^{-/-}$;*ttn.1*$^{-/-}$ (**c**,**f**) larvae at 5 dpf. Compared to wild-type fish (**a**,**d**), the muscle fiber structure was largely lost in the *ttn.1*-null zebrafish, regardless of the *srpk3* status (that is both *srpk3*$^{+/+}$;*ttn.1*$^{-/-}$ and *srpk3*$^{-/-}$;*ttn.1*$^{-/-}$). Scale bars are 25 µm. Representative images from >15 pooled fish per genotype.

**a**

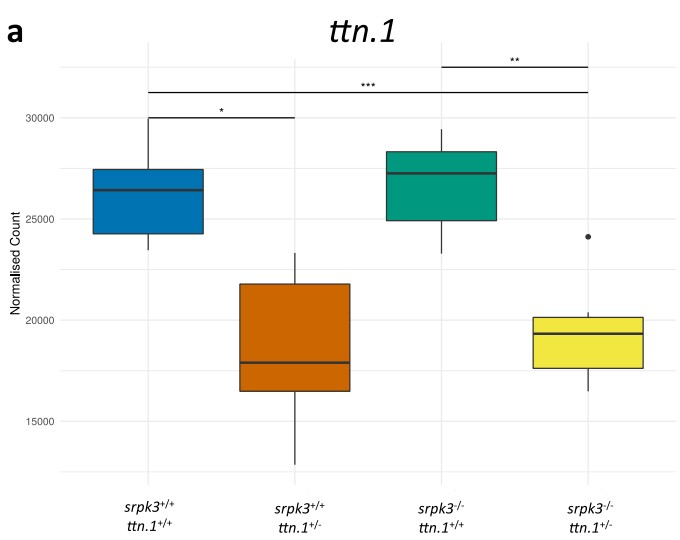

**b**

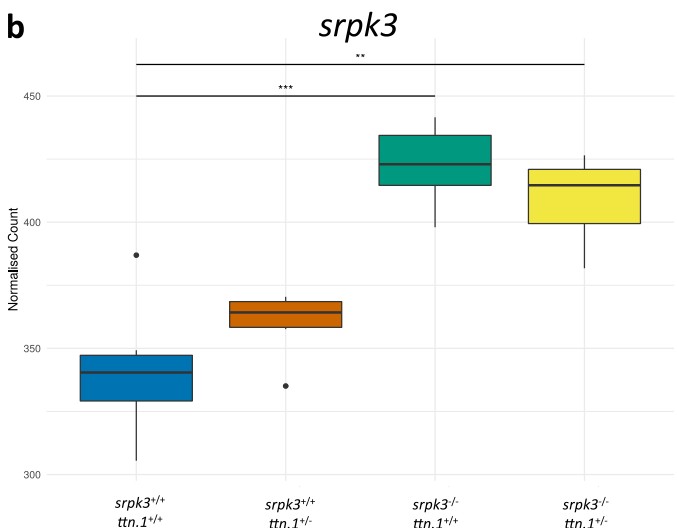

**Extended Data Fig. 9 | mRNA expression in the _srpk3_ and _ttn_ zebrafish models. a**, Transcriptome data showed that _ttn.1_ mRNA (ENSDARG00000000563) expression is equally reduced in the heterozygous (_srpk3^+/+_; _ttn.1^+/−_) and the double mutants (_srpk3^−/−_; _ttn.1^+/−_) when compared to the wild-type _ttn.1_. _P_-values: (*) _sprk3^+/+_; _ttn.1^+/−_ vs. _srpk3^+/+_; _ttn.1^+/+_ = 0.024, (**) _sprk3^−/−_; _ttn.1^+/−_ vs. _srpk3^−/−_; _ttn.1^+/+_ = 0.004, (***) _sprk3^−/−_; _ttn.1^+/−_ vs. _srpk3^+/+_; _ttn.1^+/+_ = 3.221 × 10^{−4}. **b**, _srpk3_ mRNA (ENSDARG00000005916) was upregulated in the _srpk3^−/−_ zebrafish mutants (both _srpk3^−/−_; _ttn.1^+/+_ and _srpk3^−/−_; _ttn.1^+/−_), likely

as a compensatory effect. _P_-values: (**) _sprk3^−/−_; _ttn.1^+/−_ vs. _srpk3^+/+_; _ttn.1^+/+_ = 0.005, (***) _sprk3^−/−_; _ttn.1^+/+_ vs. _srpk3^+/+_; _ttn.1^+/+_ = 5.178 × 10^{−4}. The boxes represent the first and third quartiles (25% and 75% percentile) with the center line at the median value. The whiskers extend from the hinge to the furthest value not beyond 1.5 times the interquartile range from the hinge. Any outlier values beyond 1.5 times the interquartile range are plotted as individual points. Differential expression was done using a two-sided Wald test with Benjamini-Hochberg adjustment for multiple testing. _n_ = 6 for each condition.

| GO Term | Domain | topGO pvalue | | Number of DE genes | | |
|---|---|---|---|---|---|---|
| | | srpk3$^{-/-}$; ttn.1$^{+/-}$ | srpk3$^{-/-}$; ttn.1$^{+/+}$ | srpk3$^{-/-}$; ttn.1$^{+/-}$only | Both | srpk3$^{-/-}$; ttn.1$^{+/+}$only |
| cell-matrix adhesion | BP | 1.9e−07 | 3.8e−07 | | | |
| Z disc | CC | 2.9e−07 | 2.0e−06 | | | |
| skeletal myofibril assembly | BP | 4.1e−07 | 7.9e−05 | | | |
| entrainment of circadian clock by photoperiod | BP | 2.5e−06 | 3.6e−03 | | | |
| dynactin complex | CC | 2.3e−05 | 5.8e−06 | | | |
| circadian regulation of gene expression | BP | 4.4e−05 | 3.4e−02 | | | |
| actin binding | MF | 8.4e−05 | 5.5e−05 | | | |
| skeletal muscle tissue development | BP | 1.4e−04 | 5.4e−02 | | | |
| skeletal muscle fiber development | BP | 1.5e−04 | 1.8e−05 | | | |
| superoxide metabolic process | BP | 1.5e−04 | 3.1e−05 | | | |
| inflammatory response | BP | 1.6e−04 | 4.1e−04 | | | |
| response to virus | BP | 1.9e−04 | 1.2e−03 | | | |
| response to hydrogen peroxide | BP | 2.8e−04 | 2.5e−02 | | | |
| immune response | BP | 4.4e−04 | 2.4e−02 | | | |
| sarcomere organization | BP | 5.5e−04 | 8.1e−06 | | | |
| cytokine-mediated signaling pathway | BP | 6.3e−04 | 5.9e−04 | | | |
| actinin binding | MF | 6.4e−04 | 1.9e−04 | | | |
| nucleotide-sugar biosynthetic process | BP | 8.0e−04 | 3.6e−03 | | | |
| sarcolemma | CC | 9.8e−04 | 2.1e−04 | | | |
| protein glycosylation | BP | 9.9e−04 | 4.3e−03 | | | |
| neutrophil migration | BP | 1.3e−03 | 1.8e−04 | | | |
| actin cytoskeleton | CC | 1.0e−02 | 7.1e−05 | | | |
| heart contraction | BP | 1.6e−02 | 2.2e−05 | | | |
| actin cytoskeleton organization | BP | 1.6e−02 | 7.6e−04 | | | |
| actin filament binding | MF | 2.4e−02 | 3.8e−06 | | | |
| intermediate filament | CC | 3.1e−02 | 6.7e−05 | | | |
| heart development | BP | 7.8e−02 | 1.2e−04 | | | |
| striated muscle contraction | BP | 3.0e−01 | 9.5e−04 | | | |

**Extended Data Fig. 10 | Overlap of differentially expressed (DE) genes annotated to enriched Gene Ontology (GO) terms.** The table shows the top enriched GO terms from the comparisons of *srpk3$^{-/-}$; ttn.1$^{+/-}$* and *srpk3$^{-/-}$; ttn.1$^{+/+}$* embryos to wild-type (*P* < 0.001 in one of the comparisons). The topGO *P*-value columns show the *P*-value from the enrichment test for each comparison. The dots in the number of DE genes columns represent the number of DE genes annotated to the term that appear either in *srpk3$^{-/-}$; ttn.1$^{+/-}$* alone (blue), *srpk3$^{-/-}$; ttn.1$^{+/+}$* alone (red) or in both (purple). For most GO terms, most of the DE genes causing the enrichment are shared between both lists. GO term enrichment was done using the topGO package using a one-sided Fisher's exact test without adjustment for multiple testing.

# Reporting Summary

## Statistics

For all statistical analyses, confirm that the following items are present in the figure legend, table legend, main text, or Methods section.

| n/a | Confirmed | |
|---|---|---|
| ☐ | ☒ | The exact sample size (*n*) for each experimental group/condition, given as a discrete number and unit of measurement |
| ☒ | ☐ | A statement on whether measurements were taken from distinct samples or whether the same sample was measured repeatedly |
| ☐ | ☒ | The statistical test(s) used AND whether they are one- or two-sided *Only common tests should be described solely by name; describe more complex techniques in the Methods section.* |
| ☒ | ☐ | A description of all covariates tested |
| ☐ | ☒ | A description of any assumptions or corrections, such as tests of normality and adjustment for multiple comparisons |
| ☐ | ☒ | A full description of the statistical parameters including central tendency (e.g. means) or other basic estimates (e.g. regression coefficient) AND variation (e.g. standard deviation) or associated estimates of uncertainty (e.g. confidence intervals) |
| ☐ | ☒ | For null hypothesis testing, the test statistic (e.g. *F*, *t*, *r*) with confidence intervals, effect sizes, degrees of freedom and *P* value noted *Give P values as exact values whenever suitable.* |
| ☒ | ☐ | For Bayesian analysis, information on the choice of priors and Markov chain Monte Carlo settings |
| ☒ | ☐ | For hierarchical and complex designs, identification of the appropriate level for tests and full reporting of outcomes |
| ☒ | ☐ | Estimates of effect sizes (e.g. Cohen's *d*, Pearson's *r*), indicating how they were calculated |

*Our web collection on statistics for biologists contains articles on many of the points above.*

## Software and code

Policy information about availability of computer code

| Data collection | Nikon A1R point scanning confocal microscope (Nikon, UK) Nikon Elements AR, v5.21.03 Hitachi HT7800 120 kV TEM, using a EMSIS CMOS Xarosa high-resolution camera (Hitachi, Japan) Radius software v2.0 |
|---|---|
| Data analysis | For patients' RNA seq analysis: featureCounts utility from the Subread package 2.0.0; calcNormFactors function from edgeR v3.28.1; AllelicImbalance67 software package v1.24.0 For zebrafish transcriptomics work: FastQC v0.11.9 (https://www.bioinformatics.babraham.ac.uk/projects/fastqc/); STAR v2.7.3a; DESeq2 v1.28.1; topGO v2.38.1; ClueGO v2.5.9; Cytoscape 3.9.1; R v4.2.0 (R Core Team; https://www.r-project.org/); tidyverse package v1.3.1; rMATS v4.1.2 For the WGS mouse work: BBmap suite v38.69; Varscan 2 v2.3.7; BCFTools v1.9; Variant Effect Predictor tool (Ensembl) https://www.ensembl.org/info/docs/tools/vep/index.html, https://www.ensembl.org/Tools/VEP For variant analyses: CADD v1.6 (https://cadd.gs.washington.edu/); YASARA v15.4.10; Mutation Mapper (https://www.cbioportal.org/mutation_mapper) For statistical modelling: LINKAGE package; MLINK; PSEUDOMARKER v2.0, (https://www.socscistatistics.com/tests/ztest/default2.aspx) |

For manuscripts utilizing custom algorithms or software that are central to the research but not yet described in published literature, software must be made available to editors and reviewers. We strongly encourage code deposition in a community repository (e.g. GitHub). See the Nature Portfolio guidelines for submitting code & software for further information.

## Data

Policy information about **availability of data**

All manuscripts must include a **data availability statement**. This statement should provide the following information, where applicable:
- Accession codes, unique identifiers, or web links for publicly available datasets
- A description of any restrictions on data availability
- For clinical datasets or third party data, please ensure that the statement adheres to our **policy**

> Zebrafish RNA-seq data can be accessed in the ArrayExpress database at EMBL-EBI (www.ebi.ac.uk/arrayexpress) under accession number E-MTAB-12934. Mouse WGS data and human RNA-seq data can be accessed in the Sequence Read Archive (SRA) under accession number (PRJNA1027609 and PRJNA1027754, respectively). Control frequencies and variant information were extracted from gnomAD v2.1.1 (https://gnomad.broadinstitute.org). TTN variant information was obtained from Leiden Open Variation Database (https://databases.lovd.nl/shared/genes/TTN).

## Human research participants

Policy information about **studies involving human research participants and Sex and Gender in Research.**

| | |
|---|---|
| Reporting on sex and gender | We report on 31 males and two female patients. Gender is not relevant for the study. |
| Population characteristics | Patients with congenital myopathy and rare likely damaging variants in SRPK3 were included in the study. Agreggated and individual clinical data can be found in Supplementary Tables 1 and 2. The final cohort comprised 33 patients (29 males and 2 females). Age range was 1-77 years. |
| Recruitment | Recruitment was not performed for the study |
| Ethics oversight | All clinical information and biological material used in this collaborative study was collected after obtaining written informed consent from the patients or their legal guardians. Each sequencing study was approved by their relevant Health Research Authorities, as follows: Family A: Health Research Authority, NRES Committee East of England – Hatfield (REC 06/Q0406/33). Families B, W and X: Consent approved by the French legislation (Comité de Protection des Personnes Est IV DC-2012-1693). DNA storage and usage was IRB-approved: Comités de Protection des Personnes (CPP-Est DC-2012-1693). Family C: Ethics Committee of the National Center of Neurology and Psychiatry, Japan (A2011-081). Families D, R, V, Y and Z: NRES Committee North East – Newcastle & North Tyneside 1 (REC 08/H0906/28+5). Family E: EC approval (Number S52853) by Ethical Committee Research UZ / KU Leuven. Families H and U: by the Medical Review Ethics Committee, Region Arnhem–Nijmegen, Number 2011/188. Family L: NIH, National Institute of Neurological Disorders and Stroke (NINDS), Institutional Review Board (Protocol 12-N-0095). Family M: Ethics committee of the Helsingin ja Uudenmaan sairaanhoitopiiri (HUS, statement number 195/13/03/00/11). Family N: University of Pretoria Faculty of Health Sciences Research Ethics Committee Ref 296/2019. Families O and P: Boston Children's Hospital Institutional Review Board (Protocol #03-08-128R). Family Q: Rare Genomes Project (Protocol #: 2016P001422) study, approved by the IRB at Massachusetts General Brigham. Family S: New Zealand Health and Disability Ethics Committee - approval number 20/NTB/139. Families F, G, J, K and T: tested through their respective national diagnostic health services. |

Note that full information on the approval of the study protocol must also be provided in the manuscript.

# Field-specific reporting

Please select the one below that is the best fit for your research. If you are not sure, read the appropriate sections before making your selection.

☒ Life sciences    ☐ Behavioural & social sciences    ☐ Ecological, evolutionary & environmental sciences

For a reference copy of the document with all sections, see nature.com/documents/nr-reporting-summary-flat.pdf

# Life sciences study design

All studies must disclose on these points even when the disclosure is negative.

| | |
|---|---|
| Sample size | Statistical methods were not used to determine sample size. The study cohort consisted of 33 congenital myopathy patients with rare variants in SRPK3. Disease cohorts consisted of 170, 94 and 56 individuals with confirmed diagnosis of CAPN3-, DYSF-, and ANO5-related muscular dystrophy, respectively, identified through previous sequencing projects. |
| Data exclusions | No data was excluded. |
| Replication | Zebrafish single myofiber staining images were taken from >15 pooled fish per genotype. Zebrafish EM images were taken from 3-5 pooled fish per genotype. Zebrafish transcriptomics was performed with n=6 for each condition. In vitro phosphorylation assay was performed in quadruplicate. TTN western blots were repeated twice, from the same muscle lysates. All attempts were successful. |

| Randomization | The study was not randomised. |
|---|---|
| Blinding | Investigators were blinded to sex and affected status of family members during segregation analysis. |

# Reporting for specific materials, systems and methods

We require information from authors about some types of materials, experimental systems and methods used in many studies. Here, indicate whether each material, system or method listed is relevant to your study. If you are not sure if a list item applies to your research, read the appropriate section before selecting a response.

## Materials & experimental systems

| n/a | Involved in the study |
|---|---|
| ☐ | ☒ Antibodies |
| ☐ | ☒ Eukaryotic cell lines |
| ☒ | ☐ Palaeontology and archaeology |
| ☐ | ☒ Animals and other organisms |
| ☒ | ☐ Clinical data |
| ☒ | ☐ Dual use research of concern |

## Methods

| n/a | Involved in the study |
|---|---|
| ☒ | ☐ ChIP-seq |
| ☒ | ☐ Flow cytometry |
| ☒ | ☐ MRI-based neuroimaging |

## Antibodies

| Antibodies used | Alexa Fluor® 488 Phalloidin Invitrogen A12379 (1:80)<br>mouse monoclonal anti α-actinin Sigma A7811 (1:200)<br>rabbit polyclonal anti-sarcomeric α Actinin Cell Signaling Technologies #3134, Lot #2 (1:25)<br>mouse monoclonal anti Titin Merck T9030, clone T11, ascites fluid, lot #029M4835V (1:200)<br>goat anti-mouse Alexa Fluor® 488 secondary antibody ThermoFisher A11001 (1:500)<br>goat anti-rabbit Alexa Fluor® 594 secondary antibodies ThermoFisher A11037 (1:250)<br>V5 tag ThermoFisher R960-25, SV5-Pk1 (1:5000)<br>GFP ThermoFisher A11122 (1:5000)<br>rabbit polyclonal anti-titin M10 (M10-1) as described in Hackman et al. Neuromuscul Disord 18:922–928 (2008)(1:300)<br>mouse monoclonal anti-titin M10 (11-4-3) as described in Evilä et al. Ann Neurol 75:230–240 (2014) (1:150)<br>rabbit polyclonal anti-titin Z1Z2 Myomedix, TTN-1 (1:1500)<br>mouse monoclonal anti-titin (distal I-band) Enzo Life Sciences, ALX-BC-3010-S, F146.9B9 (1:1000) |
|---|---|
| Validation | Hackman et al. Neuromuscul Disord 18:922–928 (2008)<br>Evilä et al. Ann Neurol 75:230–240 (2014) |

## Eukaryotic cell lines

Policy information about cell lines and Sex and Gender in Research

| Cell line source(s) | 293T cells (ATCC, CRL-3216) |
|---|---|
| Authentication | Cells were commercially available and were not authenticated. |
| Mycoplasma contamination | Cell line tested negative for mycoplasma contamination |
| Commonly misidentified lines<br>(See ICLAC register) | None used |

## Animals and other research organisms

Policy information about studies involving animals; ARRIVE guidelines recommended for reporting animal research, and Sex and Gender in Research

| Laboratory animals | Adult zebrafish (Danio erio) mutant lines srpk3(sa18907) and ttn.1(sa5562) were used to obtain all genotype combinations. Double mutants larvae were analysed at 5dpf. |
|---|---|
| Wild animals | No wild animals were used in this study |
| Reporting on sex | This information has not been collected |
| Field-collected samples | No field-collected samples were used in this study |
| Ethics oversight | Zebrafish were maintained in accordance with UK Home Office regulations, UK Animals (Scientific Procedures) Act 1986, under |

| Ethics oversight | project licences 70/7606 and P597E5E82. All animal work was reviewed by The Wellcome Trust Sanger Institute Ethical Review Committee. |
| --- | --- |

Note that full information on the approval of the study protocol must also be provided in the manuscript.

