## [Peer Review File · Nature Genetics]

Peer Review Information

Manuscript Title: Digenic inheritance involving a muscle specific protein kinase and the giant titin protein causes a skeletal muscle myopathy

Corresponding author name(s): Volker Straub, Ana Töpf

Reviewer Comments & Decisions:

Decision Letter, initial version:

16th July 2021

Dear Dr. Topf,

Thank you again for choosing to submit your manuscript using the Guided Open Access pilot at the Nature Portfolio. As part of this process, our editorial team has considered your paper for three of our journals with strong interest in publishing in your field: *Nature Genetics*, *Nature Communications*, and *Communications Biology*.

Your manuscript entitled "Digenic inheritance involving a muscle specific protein kinase and the giant titin protein causes a skeletal muscle myopathy" has now been reviewed by three experts in skeletal muscle diseases, human genetics, and zebrafish models, whose comments are included below and in the attached Editorial Assessment Report. As part of the Guided Open Access pilot, editors from all three journals have discussed the reviewer reports and the manuscript's suitability for our journals. After careful evaluation, our editorial recommendation is to revise the manuscript and submit back through the Guided Open Access submission portal for consideration at *Nature Genetics* using the link provided below. Provided the revisions satisfy all technical and editorial concerns, *Nature Genetics* is very interested in publishing your manuscript. Please see details in the attached Editorial Assessment Report.

In brief, for publication in *Nature Genetics*, we ask that you strengthen the molecular and phenotypic analyses of the human patient biopsies and zebrafish mutants as requested by two of the reviewers, and that you clarify aspects of the presentation throughout.

Please note that the Editorial Assessment Report is a standalone document that contains an editorial evaluation, recommendation and portable peer advice to help you navigate and interpret the reviewers' reports. It also provides guidance for adhering to best practice with regard to transparency and reproducibility, for example on the issue of sharing data. We have also included information about data accessibility and reproducibility, which we hope you find useful.

HOW TO SUBMIT

Once you are ready to submit a revised version of your manuscript, please use the link below to submit the following items as separate documents:

- Revised manuscript. Please show all changes in the manuscript text file with tracked changes or color highlighting.
- Any supplementary files.
- Point-by-point response to the reviewers' comments, reproduced verbatim. If you are unable to address specific reviewer requests or find any points invalid, please explain why.
- Cover letter to the editor, stating the journal for which you have revised.

[redacted]

Should you have any questions about the recommended journals or would like advice on the revisions, you can contact me directly and I will be happy to assist. We look forward to receiving the revised version of your manuscript.

Yours sincerely,
Kyle

--

Kyle Vogan
Editor
Guided Open Access
On behalf of the Guided OA editorial team

REVIEWER COMMENTS

Reviewer #1 (Remarks to the Author: Overall significance):

Topf et al present a novel myopathy caused by the digenic inheritance of variants in *Srpk3* and *Ttn*. They present numerous families that support this association, as well as data from other diseases to show that co-inheritance is not by chance (i.e. truncating *Ttn* variants are not found in these diseases). In addition, they provide support from zebrafish mutants, where *srpk3* $-/-$ fish only have a phenotype in the setting of *ttn* $+/-$ or $-/-$. Overall, this is an extremely compelling article presenting one of the few examples of digenic inheritance as the cause of a Mendelian appearing disease. There are some areas where additional data would strengthen the proof of this association and provide insight into disease mechanism.

Reviewer #1 (Remarks to the Author: Impact):

This study has high potential impact, as digenic inheritance may be a potential explanation for many currently unsolved cases of rare disease.

Reviewer #1 (Remarks to the Author: Strength of the claims):

There are areas where data can be obtained or added that would strengthen the claims of the manuscript:

(1) The authors appear to have access to several biopsies from the patients (and potentially unaffecteds as well?). Given the proposed function(s) of *Srpk3*, it would be informative to understand how the presence of both mutations impacts transcriptional changes in the muscle. This would be most informative related to *Ttn* (are *Ttn* transcripts altered by *Srpk3* mutation?), but also to see if *Ttn* variation alters *Srpk3* dependent RNA regulation.

(2) Also from the biopsies, the study looks at TTN levels. The westerns are not the most compelling data, as it does look like there might be some expression in at least one of the digenic patients. Is there an opportunity to more precisely define TTN quantification and post translational modification? Perhaps by using IP/MS and direct interrogation of TTN?

(3) The zebrafish data is quite sparse, making it hard to contextualize the model and understand whether it provides validation of the human data. As presented it only minimally adds to the proof.

- No information is provided about the impact of the mutation on Srpk3 RNA and protein. Presumably this is a KO but it would be important to investigate at minimum with RNA analysis.

- Only phalloidin staining is provided as phenotypic characterization. This is pretty minimal in terms of a data set. In terms of histopathology, what about proteins that illuminate other muscle structures? There are good antibodies that work in zebrafish that mark the thin filament, the Z band, and the triad (as well as the DAPC). These can be used in whole mount or on myofibers, and would provide more of a picture of the impact of the double mutation, and also enable an opportunity to better compare with the human pathology. Similarly, it would be important to show EM from the zebrafish to see whether there are analogous structural changes.

- No functional data is provided from the zebrafish. Is swim behavior impaired? How about survival? Also, what about cardiac function (heart rhythm and ventricular shortening are relatively straight forward to enumerate).

- What are TTN levels (RNA and protein) in the *srpk3* $-/-$, *ttn* $+/-$?

- Are there alterations in the global transcriptome in *srpk3* $-/-$? Are they impacted by the presence of a heterozygous *ttn* variant? How do they compare to the human biopsies?

Reviewer #1 (Remarks to the Author: Reproducibility):

The data in general appears valid and reproducible. The western blots of TTN are somewhat difficult to interpret and also lack quantification. The caveat is that these are extremely technically challenging. An alternative way (such as IP/MS) at looking at TTN levels might be helpful in this regard, as this is a

pivotal point for the study conclusions.

Reviewer #2 (Remarks to the Author: Overall significance):

I have read the manuscript by Töpf and colleagues on “Digenic inheritance involving a muscle specific protein kinase and the giant titin protein causes a skeletal muscle myopathy” with great interest.

The authors report that deleterious variants in the X-linked SRPK3 gene lead to a progressive childhood-onset skeletal muscle myopathy only when in combination with heterozygous TTN variants. Double mutant zebrafish recapitulated the clinical phenotype.

The results are original and significant, will be of interest to others in the community, and the manuscript is well written.

Reviewer #2 (Remarks to the Author: Impact):

This Reviewer thinks that this is a significant contribution to the field.

Reviewer #2 (Remarks to the Author: Strength of the claims):

The work is very convincing.

Comments:

- 1. Abstract: Are the deleterious variants all loss-of function variants? If so, I suggest adding this information.**
- 2. Please use variant or pathogenic variant instead of mutation throughout the manuscript.**
- 3. “Segregation analyses (in 16 families) showed that SRPK3 variants were always inherited from an unaffected mother, with none being de novo.” Would it be possible to test if the variants were de novo in the unaffected mothers?**
- 4. How did the authors define “partially skewed X-inactivation”?**
The authors pointed out, that “Interestingly, individual VII:5, a female SRPK3 carrier showing a pronounced skewed X-inactivation pattern (3:97) but no TTN

variant, was clinically unaffected. This Reviewer wonders why this is interesting.

5. “Also unaffected was individual NI:2,.....” Please revise. This Reviewer understood that all females with co-segregating SRPK3 and TTN variants, except for the two carriers in family U are unaffected.

6. Results on the genetic background of the Srpk3 KO mouse are interesting, but descriptive and it remains unclear whether the Ttn variants contribute to the observed phenotype.

Discussion:

7. Line 289: “Six females also carried both...”. Fig. 1C shows nine females with both SRPK3 and TTN variants.

8. While numbers are still small, X-inactivation results are repeated rather instead of being discussed.

9. Do the authors have any evidence that SRPK3 and titin are complex partners?

10. Extended data Fig.2: p.Glu455Ly is not a SRPK3 variant of Extended Table 1.

11. Again, numbers are still small. Nevertheless, this Reviewer wonders whether it would make sense mentioning that all SRPK3 missense variants identified so far are located in the kinase domains.

Reviewer #3 (Remarks to the Author: Overall significance):

In this manuscript the authors present an example of digenic inheritance related to a potential interaction between SRPK3, a X-linked kinase, and TTN, the giant muscle protein. They analyzed a cohort of patients with deleterious variants in SRPK3 in combination with heterozygous variants in TTN. To support their hypothesis of digenic inheritance they used zebrafish as animal model by creating double carrier fish of the srpk3 and ttn.1 variants.

This new finding is very interesting, but the data are not very well presented, therefore I have a number of major concerns.

Line 165: the authors write about NMD (nonsense mediated mRNA decay). Is this a hypothesis? In the Extended Fig. 1 there is no proof related to it. A qPCR would be nice to prove the downregulation of mRNA levels.

Line 215: the authors describe for the patient NII:1 elevated levels of CK. Why do the authors report this fact? Could this influence the phenotype? Was this condition in the patient related only to the specific moment/period the patient was tested or is this a standard condition of the patient?

Line 216 to 224: the authors describe too roughly the Fig 2A and without having a look on the figure legend it is impossible to understand to which patient the description relates. It would be nice if the authors briefly mention in the text whether the finding occurs in all the patients or only in few. Please rephrase this part?

Line 231: Based on which criteria the authors decided to perform the western blot using only those 4 patients? And why didn't they consider to perform the western blot analysis also using samples from patients with a myopathic phenotype (families M and V)?

Fig. 2C: The quality of the WB is not adequate. Please provide better quality pictures? Related to the N-term titin blot, the authors describe the presence of a clear titin band in sample DI:1. Actually the band is not really clear and there are a lot of smears for the N-term and I-band blots.

The authors should consider to perform a new western blot decreasing the percentage of the gel (to 1.8-2.5%) for a better quality (as published from Swist et al. 2020 doi: 10.1038/s41467-020-18131-2).

The authors performed the staining in zebrafish embryos at 5days post fertilization. It would be nice to know for how long the embryos that display the muscle fiber disruption survive. Do they survive to adulthood? Is motility affected in these embryos? Is sarcomerogenesis affected early on (24hpf) or is this degeneration of muscle fibres over time (e.g. day2-5)? Is heart function also affected in the double mutation carriers? The description of muscle fibres disruption is superficial and should be expanded by time-course analyses, by the use of more sarcomeric markers and electron microscopy imaging.

Lines 277 to 284: srpk3 KO mice show a muscle phenotype and it is not known if ttn variants are contributing to the phenotype. On the other hand, the srpk3

mutant zebrafish line does not show any muscle phenotypes. Hence, it would be interesting to overexpress the specific titin missense variants (detected in mice) in zebrafish (by Tol2 transgenesis) and to analyze a potential phenotype.

Lines 322 to 329: It would be nice if the authors could show SRPK3 and RBM20 protein levels by western blotting (using the same samples as used for Fig. 2C). Is RBM20 mislocalizing (immunostainings)?

Minor:

End of the line 267: there is an “and” that is repeated also after the parenthesis in line 268.

Author Rebuttal to Initial comments

REVIEWER COMMENTS

Reviewer #1

Reviewer #1 (Remarks to the Author: Overall significance):

Topf et al present a novel myopathy caused by the digenic inheritance of variants in *Srpk3* and *Ttn*. They present numerous families that support this association, as well as data from other diseases to show that co-inheritance is not by chance (i.e. truncating *Ttn* variants are not found in these diseases). In addition, they provide support from zebrafish mutants, where *srpk3* $-/-$ fish only have a phenotype in the setting of *ttn* $+/-$ or $-/-$. Overall, this is an extremely compelling article presenting one of the few examples of digenic inheritance as the cause of a Mendelian appearing disease. There are some areas where additional data would strengthen the proof of this association and provide insight into disease mechanism.

Reviewer #1 (Remarks to the Author: Impact):

This study has high potential impact, as digenic inheritance may be a potential explanation for many currently unsolved cases of rare disease.

We thank Reviewer 1 for the positive comments about the impact and significance of our study.

Reviewer #1 (Remarks to the Author: Strength of the claims):

There are areas where data can be obtained or added that would strengthen the claims of the manuscript:

- (1) The authors appear to have access to several biopsies from the patients (and potentially unaffecteds as well?). Given the proposed function(s) of *Srpk3*, it would be informative to understand how the presence of both mutations impacts transcriptional changes in the muscle. This would be most informative related to *Ttn* (are *Ttn* transcripts altered by *Srpk3* mutation?),

but also to see if Ttn variation alters *Srpk3* dependent RNA regulation.

We thank the reviewer for this relevant point. We had indeed obtained transcriptome data from a small number (n=3) of muscle biopsies of the *SRPK3/TTN* patients available at the time. Unfortunately, this had been carried out before we discovered the co-segregating *TTN* variants and the digenic mode of inheritance seen in these patients. We had performed standard poly(A) mRNA enrichment sequencing which is known not to be ideal for the analysis of large transcripts such as full length *TTN* mRNA due to being strongly affected by 3' positional bias. Despite this, we did re-analyse the available transcriptome data and can now show that *SRPK3* mRNA levels were significantly reduced in three of the patients with *SRPK3* truncating variants [line 182, Extended Data Fig. 2]. Likewise, *TTN* mRNA levels were also significantly reduced in these *SRPK3/TTN* patients [line 263, Extended Data Fig. 5]. We have looked extensively at the effect of the presence of the *SRPK3* variants on *TTN* transcripts, in particular abhorrent splicing, but we have not been able to detect a clear signal of this. Similarly, the RNA sequencing data were unfortunately too noisy to detect differences in transcriptional signatures in the patients.

Conversely, meaningful results were gained from RNA sequencing data from the zebrafish model. We obtained transcriptomes from four genotypes: wild type, *ttn* heterozygous, *srpk3* null and the double mutant (*srpk3*^{-/-};*ttn*^{+/-}). Here we also clearly saw reduced expression of *ttn* mRNA in the heterozygous *ttn* zebrafish yet these levels were not significantly affected by the addition of the *srpk3* mutation [Lline 343, Extended Data Fig. 9A], highlighting that the effect in the double mutant is a downstream post-transcriptional effect. Unexpectedly, *srpk3* mRNA levels were upregulated both in the *srpk3* null and the double mutant (*srpk3*^{-/-};*ttn*^{+/-}), likely as the result of a compensatory effect [line 348, Extended Data Fig. 9B]. We have also looked at the genes and pathways differentially expressed in the double mutants in comparison with the other relevant genotypes, and noted that transcriptional signals in the *srpk3*-null (*srpk3*^{-/-};*ttn*.1+/+) zebrafish and the double-mutant (*srpk3*^{-/-};*ttn*.1+/-) were similar. Expression of genes involved in myofibril and actin cytoskeletal organization, and skeletal muscle tissue development was affected. We have included these findings in the revised version of the manuscript [lines 350-363].

(2) Also from the biopsies, the study looks at TTN levels. The westerns are not the most compelling data, as it does look like there might be some expression in at least one of the digenic patients. Is there an opportunity to more precisely define TTN quantification and post translational modification? Perhaps by using IP/MS and direct interrogation of TTN?

We thank the reviewer for this very relevant suggestion. Unfortunately, precise TTN quantification was not possible as we did not have sufficient biopsy material from the patients. We nevertheless repeated the western blots (for the N-terminal and I-band antibodies) with the remaining muscle tissue lysates. The new WB clearly shows lack or reduced expression of full-length titin in the *SRPK3/TTN* patients, which confirms our previous results. The new images are neater and easier to interpret; we have now included these as a new separate figure [Fig. 3].

(3) The zebrafish data is quite sparse, making it hard to contextualize the model and understand whether it provides validation of the human data. As presented it only minimally adds to the proof.

- No information is provided about the impact of the mutation on *Srpk3* RNA and protein. Presumably this is a KO but it would be important to investigate at minimum with RNA analysis.

We thank the reviewer for highlighting this. The zebrafish *srpk3* KO (*srpk3^{sa18907}*) carries a homozygous essential splice site mutation affecting the donor site of exon 15 of *srpk3*. This mutation results in aberrant splicing leading to partial retention of intron 15 or loss of exon 15, as shown by the cDNA analysis and gel electrophoresis. As mentioned, the zebrafish transcriptome data shows transient upregulation of *srpk3* mRNA, as a result of a compensatory mechanism. We have added this information in the main text [lines 306 and 348] and included a new figure [Extended Data Fig. 7].

- Only phalloidin staining is provided as phenotypic characterization. This is pretty minimal in terms of a data set. In terms of histopathology, what about proteins that illuminate other muscle structures? There are good antibodies that work in zebrafish that mark the thin filament, the Z band, and the triad (as well as the DAPC). These can be used in whole mount or on myofibers, and would provide more of a picture of the impact of the double mutation, and also enable an opportunity to better compare with the human pathology. Similarly, it would be important to show EM from the zebrafish to see whether there are analogous structural changes.

We thank the reviewer for suggesting a better characterisation of the zebrafish muscle morphology. We have now included a Z-band marker (α -actinin) for whole mount zebrafish and have also carried out immunoanalysis of single myofibers with a titin antibody, as well as electron microscopy for the most relevant genotypes. Even though myotomes were properly formed, muscle fibers were distorted and disintegrated to a variable extent in the double mutant (*srpk3^{-/-}; ttn.1^{+/-}*). The EM very clearly shows the effect of the addition of the *ttn* heterozygous variant in the *srpk3* null background, in particular the structural disorganization around the Z-disk. In addition, the T11 antibody immunoanalysis showed a marked reduction in titin expression in the *ttn* heterozygous zebrafish, only upon loss of *srpk3* (i.e. *srpk3*-null background). We have thus compiled a new figure [Fig. 4] for a much thorough phenotypic characterization of the zebrafish and included the description in the text [lines 304-339]. We have also included images for the *ttn*-null zebrafish as a separate figure [Extended Data Fig. 8].

- No functional data is provided from the zebrafish. Is swim behaviour impaired? How about survival? Also, what about cardiac function (heart rhythm and ventricular shortening are relatively straight forward to enumerate).

Thanks for highlighting this point. Double mutants (*srpk3^{-/-}; ttn.1^{+/-}*) are not able to fill the swim bladder and therefore do not survive to adulthood. Heart and cardiac function are normal, there was no evidence of pericardiac edema or other phenotypes usually associated with heart defects in zebrafish larvae. This information has now been included in the manuscript [lines 313-318].

- What are TTN levels (RNA and protein) in the *srpk3* $-/-$, *ttn* $+/-$?

Transcriptomics data shows that both the single (*ttn* $^{+/-}$) and double (*srpk3* $^{-/-}$; *ttn* $^{+/-}$) titin mutants have reduced *ttn* mRNA levels compared to wild type fish. Interestingly, at the protein level, the isolated single fiber immunostaining clearly shows that titin protein expression in the single heterozygous (*ttn* $^{+/-}$) is comparable to wild type controls but is remarkably reduced in the double mutants suggesting that the effect in the double mutant is at the post-transcriptional or post-translational level. This has been discussed at length in the manuscript [lines 332-338 and 426-423 and Extended Data Fig. 9A,B].

- Are there alterations in the global transcriptome in *srpk3* $-/-$? Are they impacted by the presence of a heterozygous *ttn* variant? How do they compare to the human biopsies?

Many thanks for bringing up this very relevant point. We have now obtained global transcriptome data for the relevant zebrafish genotypes. Expression analysis shows that 128 genes are differentially expressed in the heterozygous *ttn* mutant zebrafish, compared to wild type. In the *srpk3*-null, however, a much larger number of genes is de-regulated ($n=572$), and this is even more pronounced with the addition of a *ttn* variant ($n=794$). GO enrichment analysis showed that these transcripts belong mainly to the sarcomere and muscle structure development. These results are described in the text [lines 350-363] and shown in new figures [Fig. 5 and Extended Data Figure 10]. The full list of genes differentially expressed in the various genotypes is also provided [Extended Data Table 3]. Unfortunately, similar analysis of the available *SRPK3/TTN* patients did not have enough power to detect this signal.

Reviewer #1 (Remarks to the Author: Reproducibility):

The data in general appears valid and reproducible. The western blots of TTN are somewhat difficult to interpret and also lack quantification. The caveat is that these are extremely technically challenging. An alternative way (such as IP/MS) at looking at TTN levels might be helpful in this regard, as this is a pivotal point for the study conclusions.

We agree with this Reviewer that titin quantification is an important point for the study, and also that titin western blots are indeed extremely technically challenging. Unfortunately, as mentioned before, no patients' biological material was available for further protein analyses, and therefore trying an alternative assay was not possible. We nevertheless repeated the western blots and think that the new results are clearer and confirm our previous findings regarding the abnormal full-length titin expression. We now show these results separately, in Fig. 3.

Reviewer #2

Reviewer #2 (Remarks to the Author: Overall significance):

I have read the manuscript by Töpf and colleagues on “Digenic inheritance involving a muscle specific protein kinase and the giant titin protein causes a skeletal muscle myopathy” with great interest.

The authors report that deleterious variants in the X-linked *SRPK3* gene lead to a progressive childhood-onset skeletal muscle myopathy only when in combination with heterozygous *TTN* variants. Double mutant zebrafish recapitulated the clinical phenotype.

The results are original and significant, will be of interest to others in the community, and the manuscript is well written.

Reviewer #2 (Remarks to the Author: Impact):

This Reviewer thinks that this is a significant contribution to the field.

We thank the Reviewer for the positive comment about the impact and significance of the study.

Reviewer #2 (Remarks to the Author: Strength of the claims):

The work is very convincing. Comments:

1. Abstract: Are the deleterious variants all loss-of function variants? If so, I suggest adding this information.

Many thanks for this suggestion. We expect these *SRPK3* variants to be loss-of-function although we do not have conclusive data on this; we have therefore used the term ‘deleterious’ instead [line 133].

2. Please use variant or pathogenic variant instead of mutation throughout the manuscript.

This has been changed throughout the manuscript.

3. “Segregation analyses (in 16 families) showed that *SRPK3* variants were always inherited from an unaffected mother, with none being de novo.” Would it be possible to test if the variants were de novo in the unaffected mothers?

We were not able to test this, as DNA from previous generations was not collected. In any case, we trust this information would not change the interpretation of the genetic findings in our *SRPK3/TTN* patients.

4. How did the authors define “partially skewed X-inactivation”?

Thanks for bringing attention to this; we have now removed the word ‘partially’. We have stated the exact X-inactivation ratios in the legend of Fig. 1 and in the Extended Data Table 1.

5. The authors pointed out, that “Interestingly, individual VII:5, a female *SRPK3* carrier showing a pronounced skewed X-inactivation pattern (3:97) but no *TTN* variant, was clinically unaffected. This Reviewer wonders why this is interesting.

Many thanks for asking us to clarify this point, which, we agree, was confusing. Individual VII:5 (*) is an unaffected female carrying only a *SRPK3* variant. If *SRPK3* variants *on their own* were sufficient to cause disease, then, it could be expected that *SRPK3* carrier females showing skewed X-inactivation, would present a (milder) myopathic phenotype. However, this is not the case in this female, supporting the digenic inheritance model where the additional *TTN* variant is needed to cause a myopathy. We have rephrased this section to make the statement clearer [line 217-225].

(*) Please note that due to the two additional *SRPK3/TTN* patients included in the cohort, this patient is now XII:5.

6. “Also unaffected was individual NI:2,.....” Please revise. This Reviewer understood that all females with co-segregating *SRPK3* and *TTN* variants, except for the two carriers in family U are unaffected.

The Reviewer’s understanding is correct: all females with co-segregating *SRPK3* and *TTN* variants, except for the two carriers in family U (showing skewed X-inactivation) (*) are unaffected. As before, we have rephrased this section to make the statement clearer [lines 217-225].

(*) Please note that due to the two additional *SRPK3/TTN* patients included in the cohort, this family is now family W.

7. Results on the genetic background of the *Srp3* KO mouse are interesting, but descriptive and it remains unclear whether the *Ttn* variants contribute to the observed phenotype.

Many thanks for this relevant comment.

We unfortunately are not able to say at this point whether the *Ttn* variants contribute to the observed phenotype in the mouse or not. We nevertheless included these results as we considered that, given the phenotype of the *Srp3* KO mouse, interrogating its genome was essential. It might well be that in contrast to humans, loss of function of *SRPK3* in both the zebrafish and the mouse led to a mild-moderate skeletal myopathy. Discrepancies between murine models and their human counterpart are not uncommon. For example, in some dystroglycanopathies the mouse model presents a more severe phenotype, whereas in others is a lot milder (e.g. for LGMD-R9, *FKRP*). We have expanded this in the discussion [lines 515-520].

Discussion:

8. Line 289: “Six females also carried both...” Fig. 1C shows nine females with both *SRPK3* and *TTN* variants.

Thanks for highlighting this. The Reviewer is correct that Fig. 1C shows nine females with both

SRPK3 and *TTN* variants. Line 289, however, referred to those females where X-inactivation data was available. We have rephrased this to make it clearer [line 405].

9. While numbers are still small, X-inactivation results are repeated rather instead of being discussed.

Thanks for pointing this out. We have rephrased this section [lines 401-408].

10. Do the authors have any evidence that *SRPK3* and titin are complex partners?

Many thanks for this very important question. There is no indication, functional or from the literature, that *SRPK3* and titin are complex partners. However, we can now show for the first-time, that *SRPK3* phosphorylates (and thus likely activates) the splicing factor *RBM20*. Given that *RBM20* is well-known for regulating titin splicing, the *SRPK3*-mediated *RBM20* phosphorylation is strong evidence of the (indirect) link between *SRPK3* and titin. We have included this finding as a new section in the manuscript [lines 375-395].

11. Extended data Fig.2: p.Glu455Lys is not a *SRPK3* variant of Extended Table 1.

Thanks for pointing out this typo. The variant is now included in Table 1 and Extended Data Table 1.

12. Again, numbers are still small. Nevertheless, this Reviewer wonders whether it would make sense mentioning that all *SRPK3* missense variants identified so far are located in the kinase domains.

Many thanks for this suggestion. We have now added this information in the text [line 184].

Reviewer #3

Reviewer #3 (Remarks to the Author: Overall significance): In this manuscript the authors present an example of digenic inheritance related to a potential interaction between *SRPK3*, a X-linked kinase, and *TTN*, the giant muscle protein. They analyzed a cohort of patients with deleterious variants in *SRPK3* in combination with heterozygous variants in *TTN*. To support their hypothesis of digenic inheritance they used zebrafish as animal model by creating double carrier fish of the *srpk3* and *ttn.1* variants. This new finding is very interesting, but the data are not very well presented, therefore I have a number of major concerns.

Line 165: the authors write about NMD (nonsense mediated mRNA decay). Is this a hypothesis? In the Extended Fig. 1 there is no proof related to it. A qPCR would be nice to prove the downregulation of mRNA levels.

Many thanks for pointing this out. We unfortunately do not have biomaterial available for further experiments but have now included mRNA expression data levels for three patients with *SRPK3*

truncating variants for whom transcriptome data was available. We show that for these, *SRPK3* expression are significantly lower than in controls [line 181-183] and Extended Data Fig. 2].

Line 215: the authors describe for the patient NII:1 elevated levels of CK. Why do the authors report this fact? Could this influence the phenotype? Was this condition in the patient related only to the specific moment/period the patient was tested or is this a standard condition of the patient?

Many thanks for allowing us to clarify this point. Serum creatine kinase levels are an indirect marker of muscle damage but do not *per se* influence the phenotype. We mentioned the CK levels in NII:1 (*), as this is the only patient in the *SRPK3/TTN* cohort with a significantly elevated value. As queried by the Reviewer, we made enquiries and now know that the CK values in this patient were consistently high, ranging between 1000-3000 U/l, although the cause of this is unknown. We have added this information [line 238 and Extended Data Table 2 legend].

(*) Please note that due to the two additional *SRPK3/TTN* patients included in the cohort, this patient is now PII:1.

Line 216 to 224: the authors describe too roughly the Fig 2A and without having a look on the figure legend it is impossible to understand to which patient the description relates. It would be nice if the authors briefly mention in the text whether the finding occurs in all the patients or only in few. Please rephrase this part?

Many thanks for highlighting this very valid point. We have rephrased this section and made clear the number of patients in which each specific histopathological finding was observed [lines 239-243]. We have also clarified this in Fig. 2 legend.

Line 231: Based on which criteria the authors decided to perform the western blot using only those 4 patients? And why didn't they consider to perform the western blot analysis also using samples from patients with a myopathic phenotype (families M and V)?

Many thanks for allowing us to clarify this important point. The western blot analysis was performed on the biomaterial (i.e. muscle biopsy) that was available to us at the time (rather than chosen based on any criteria). The WB shown in Fig. 3 includes five samples: four *SRPK3/TTN* myopathic patients (DII:1, LII:1, VIII:1, VII:3 and WII:3) and one unaffected relative (DI:1, father of DII:1) who only carries the *TTN* truncating variant (but not the *SRPK3* change).

However, please note that *all SRPK3/TTN* patients described in the manuscript present with a myopathic phenotype, and not just those from families M and V (*), as implied by this Reviewer's comment. Families M and V present both myopathy *and* cardiomyopathy, but unfortunately biomaterial was not

available for either of them.

(*) Please note that due to the two additional *SRPK3/TTN* patients included in the cohort, this family is now family X.

Fig. 2C: The quality of the WB is not adequate. Please provide better quality pictures? Related to the N-term titin blot, the authors describe the presence of a clear titin band in sample DI:1. Actually the band is not really clear and there are a lot of smears for the N-term and I-band blots. The authors should consider to perform a new western blot decreasing the percentage of the gel (to 1.8-2.5%) for a better quality (as published from Swist et al. 2020 doi: 10.1038/s41467-020-18131-2).

Many thanks for pointing this out. We agree with the Reviewer that the quality of the western blots was not ideal. We have now carried out new N-terminal and I-band blots with the remaining muscle lysates we had (as unfortunately fresh muscle biopsies were not available). We think these are neater and showed no smear. We have replaced the images and created a separate, much easier to interpret, Fig. 3.

The authors performed the staining in zebrafish embryos at 5days post fertilization. It would be nice to know for how long the embryos that display the muscle fiber disruption survive. Do they survive to adulthood? Is motility affected in these embryos?

Many thanks for raising this question. Double mutants are not able to fill the swim bladder due to reduce motility and therefore do not survive to adulthood. We have added this information in the manuscript [lines 315-318]. Of note, a recent pre-print describes adult *Srpk3* null zebrafish as viable (Kim, C.H. et al. Eye movement defects in KO zebrafish reveals *SRPK3* as a causative gene for an X-linked intellectual disability. Res Sq (2023))

Is sarcomerogenesis affected early on (24hpf) or is this degeneration of muscle fibres over time (e.g. day2-5)?

Sarcomerogenesis does not seem to be affected early on; double mutants have properly patterned sarcomeres, suggesting that the muscle degeneration is over time.

Is heart function also affected in the double mutation carriers?

Heart morphology and cardiac function appear normal in the *srpk3/ttn* double mutants, but this has not been formally tested. We have added this information in the manuscript [lines 316-317].

The description of muscle fibres disruption is superficial and should be expanded by time-course analyses, by the use of more sarcomeric markers and electron microscopy imaging.

Many thanks for highlighting this. We have expanded the description of the zebrafish muscle pathology by using additional sarcomeric markers both in whole mount and isolated single fiber immunostainings. We also performed electron microscopy for the relevant genotypes. Even though myotomes were properly formed, muscle fibers were distorted and disintegrated to a variable extent in the double mutant (*srpk3*^{-/-}; *ttn.1*^{+/-}). The EM very clearly shows the effect of the addition of the *ttn* variant in the *srpk3* null background, in particular the structural disorganization around the Z-disk. In addition, the T11 antibody immunoanalysis showed a marked reduction in titin expression in the *ttn* heterozygous zebrafish, only when together with the *srpk3* null background. We have described these findings and discussed them at length in the revised version of the manuscript [lines 318-339].

Lines 277 to 284: *srpk3* KO mice show a muscle phenotype and it is not known if *ttn* variants are contributing to the phenotype. On the other hand, the *srpk3* mutant zebrafish line does not show any muscle phenotypes. Hence, it would be interesting to overexpress the specific titin missense variants (detected in mice) in zebrafish (by Tol2 transgenesis) and to analyze a potential phenotype.

We agree with the Reviewer that this would be a very interesting experiment. It is, however, outside the scope of this manuscript, as also acknowledged by the Editor. In any case, we want to clarify that the *srpk3* mutant zebrafish line is in fact very mildly affected but not indistinguishable from the wild type. The *srpk3* KO mice is indeed more severe, but it should be noted that discrepancies between murine models and their human counterpart are not uncommon.

Lines 322 to 329: It would be nice if the authors could show SRPK3 and RBM20 protein levels by western blotting (using the same samples as used for Fig. 2C). Is RBM20 mislocalizing (immunostainings)?

We thank the Reviewer for this suggestion, and we agree this would be very interesting; unfortunately, we had no human biomaterial left to perform this experiment. We do know, however, thanks to the

transcriptomics profile of the zebrafish model, that *rbm20* mRNA is upregulated both in the *srpk3* null and the double mutant (*srpk3*^{-/-}; *ttn.1*^{+/-}), suggesting this is a result of the lack of active RBM20 due to *srpk3* deficiency. We have included this in the manuscript [lines 393-395].

Minor: End of the line 267: there is an “and” that is repeated also after the parenthesis in line 268.

Corrected.

Decision Letter, first revision:

Dear Volker,

Thank you once again for choosing to submit your manuscript using the Guided Open Access pilot at the Nature Portfolio.

I am delighted to say that your revised manuscript entitled "Digenic inheritance involving a muscle specific protein kinase and the giant titin protein causes a skeletal muscle myopathy" has been seen by the original referees, and in light of their advice, we are happy, in principle, to publish your paper in Nature Genetics as an Article pending final revisions to address the referees' remaining requests and to comply with our editorial formatting and style requirements.

We are now performing detailed checks on your paper, and we will send you a checklist with our editorial and formatting requirements in approximately 1-2 weeks. Please do not upload the final materials or make any revisions until you receive this additional information from us.

OPEN ACCESS

All articles Published via Guided Open Access are made freely accessible upon publication under a [CC BY license](http://creativecommons.org/licenses/by/4.0) (Creative Commons Attribution 4.0 International License). This license allows maximum dissemination and re-use of open access materials and is preferred by many research funding bodies.

As part of the Guided Open Access pilot, the top-up article-processing charge (APC) of €2600 is now due. More information about the Open Access fees associated with the Guided Open Access pilot can be found [here](https://www.nature.com/nature-portfolio/open-access/guided-open-access).

One of our Editorial Assistants will be in touch shortly to collect the forms required for creating an invoice for the payment of these fees, as well as the appropriate Open Access License to Publish form, which is necessary in order to publish your work. If you have any questions about this, or any of our open access policies, please do hesitate to contact them at guidedOA@nature.com.

Thank you again for your interest in Guided Open Access, and please do not hesitate to contact me if you have any questions.

Sincerely,
Kyle

Kyle Vogan
Editor
Guided Open Access

Reviewers comments:

Reviewer #1 (Remarks to the Author: Overall significance):

The authors convincingly demonstrate digenic inheritance of SPRK/TTN variants as a cause of congenital myopathy. This is a novel and impactful finding that will be of interest to the broad community of human geneticists.

Reviewer #1 (Remarks to the Author: Impact):

This is a study of high impact. Digenic inheritance is likely to account for many of the currently unsolved cases of rare disease where previously we have been assuming that the cause is mono genetic.

Reviewer #1 (Remarks to the Author: Strength of the claims):

The data strongly support the conclusions. The authors were very receptive to reviewer comments, and have provided additional data to further justify their hypotheses.

1. One new element of the resubmission is the emphasis on RBM20 phosphorylation as a potential disease mechanism. Given that RBM20 is known to influence specific aspects of TTN RNA processing, it would be ideal if the authors looked at this with the data that they already generated. In particular, they

should be able to look at the regions regulated by RBM20 in both their human and zebrafish RNAseq data.

2. Also related to the RNAseq - could the authors please precisely define and show the changes to the sprk transcript in the -/- zebrafish? RNAseq should better show all the possible changes (which are often multiple with splice mutants) to the transcript.

3. Related to the RNAseq in zebrafish, I agree that increased levels of sprk could be compensatory, and are also something frequently observed in zebrafish mutants. However, this transcriptional feedback does complicate interpretation of changes in other genes. For example, there is a lot of emphasis on the RBM20 transcriptional increase. However, this is not supported by protein data, so it is really hard to say if RBM20 is actually increased or not. In the absence of more evidence, I would de-emphasize this point in the discussion.

Reviewer #1 (Remarks to the Author: Reproducibility):

Data are rigorous. Statistical methods seem appropriate. There were concerns on the initial submission regarding TTN westerns but these have been well addressed by the resubmission.

Reviewer #2 (Remarks to the Author: Overall significance):

This reviewer agrees with the authors' answers and the corresponding changes in the manuscript.

The new results on SRPK3 and RBM20 are commendable, but do not show that SRPK3 phosphorylates RBM20. Further sophisticated experiments would have to be performed to make this statement. Is it known whether SRPK3 and RBM20 are complex partners?

The results on this (in vitro and transiently expressed plasmids) are over-interpreted and the statements should be weakened.

Typo in line 392: CKL1 should be CLK1.

Reviewer #3 (Remarks to the Author: Overall significance):

The results are original and significant, will be of interest to others in the community, and the manuscript is well written and well referenced to the previous existing literature.

Reviewer #3 (Remarks to the Author: Impact):

This paper will give a significant contribution to the field

Reviewer #3 (Remarks to the Author: Strength of the claims):

In the revised version of the manuscript, the authors nicely addressed every comment and perplexity.

Author Rebuttal, first revision:

Point-by-point response to reviewers' comments

Reviewer #1 (Remarks to the Author: Overall significance):

The authors convincingly demonstrate digenic inheritance of SPRK/TTN variants as a cause of congenital myopathy. This is a novel and impactful finding that will be of interest to the broad community of human geneticists.

We thank Reviewer #1 for the positive comments.

Reviewer #1 (Remarks to the Author: Impact):

This is a study of high impact. Digenic inheritance is likely to account for many of the currently unsolved cases of rare disease where previously we have been assuming that the cause is mono genetic.

We thank again Reviewer #1 for the positive comments about the impact and significance of our study.

Reviewer #1 (Remarks to the Author: Strength of the claims):

The data strongly support the conclusions. The authors were very receptive to reviewer comments, and have provided additional data to further justify their hypotheses.

We are very pleased to read that Reviewer #1 found the additional data and changes to the manuscript satisfactory.

1. One new element of the resubmission is the emphasis on RBM20 phosphorylation as a potential

disease mechanism. Given that RBM20 is known to influence specific aspects of TTN RNA processing, it would be ideal if the authors looked at this with the data that they already generated. In particular, they should be able to look at the regions regulated by RBM20 in both their human and zebrafish RNAseq data.

We agree with Reviewer #1, and in fact, we had already interrogated the RNAseq data previously generated. Unfortunately, we have been unable to detect a signal of abnormal RBM20-regulated *TTN* splicing both in human and zebrafish. As mentioned in our previous letter, this is likely due to an unfortunate methodological error in the early stages of the study. The standard poly(A) mRNA enrichment sequencing performed made the analysis of large transcripts such as full length *TTN* mRNA, unreliable due to being strongly affected by 3' positional bias.

2. Also related to the RNAseq - could the authors please precisely define and show the changes to the *srpk* transcript in the -/- zebrafish? RNAseq should better show all the possible changes (which are often multiple with splice mutants) to the transcript.

We thank Reviewer #1 for this suggestion. We have now created a Sashimi plot for the *srpk3* transcript in the *srpk3* null zebrafish. This confirms what was seen in the PCR amplification from mRNA: partial intron retention as well as exon 15 skipping. We have included the diagram as Extended Data Fig. 7b.

3. Related to the RNAseq in zebrafish, I agree that increased levels of *srpk* could be compensatory, and are also something frequently observed in zebrafish mutants. However, this transcriptional feedback does complicate interpretation of changes in other genes. For example, there is a lot of emphasis on the RBM20 transcriptional increase. However, this is not supported by protein data, so it is really hard to say if RBM20 is actually increased or not. In the absence of more evidence, I would de-emphasize this point in the discussion.

Thank you for this valid point. We have weakened the discussion about the increased RBM20 mRNA expression.

Reviewer #1 (Remarks to the Author: Reproducibility):

Data are rigorous. Statistical methods seem appropriate. There were concerns on the initial submission regarding TTN westerns but these have been well addressed by the resubmission.

We thank Reviewer #1 for the positive comment.

Reviewer #2 (Remarks to the Author: Overall significance):

This reviewer agrees with the authors' answers and the corresponding changes in the manuscript.

We are very pleased to read that Reviewer #2 found our answers and changes to the manuscript satisfactory.

The new results on SRPK3 and RBM20 are commendable, but do not show that SRPK3 phosphorylates RBM20. Further sophisticated experiments would have to be performed to make this statement. Is it known whether SRPK3 and RBM20 are complex partners?

The results on this (in vitro and transiently expressed plasmids) are over-interpreted and the statements should be weakened.

Thank you for raising this very valid point. No, it is not known whether SRPK3 and RBM20 are complex partners, but the fact that another SR kinase, SRPK1, does phosphorylate the RSRSP stretch of RBM20 (Sun et al, 2022) is encouraging. In any case, as suggested, we have toned down the interpretation of the phosphorylation assay and made clear that the link between SRPK3 and RBM30 is still speculative at this point.

Typo in line 392: CKL1 should be CLK1.

Thanks for spotting this, it has been corrected.

Reviewer #3 (Remarks to the Author: Overall significance):

The results are original and significant, will be of interest to others in the community, and the manuscript is well written and well referenced to the previous existing literature.

Reviewer #3 (Remarks to the Author: Impact):

This paper will give a significant contribution to the field

Reviewer #3 (Remarks to the Author: Strength of the claims):

In the revised version of the manuscript, the authors nicely addressed every comment and perplexity.

We thank Reviewer #3 for the very positive feedback. We are very pleased to read that the current version of the manuscript addressed all previous comments.

Final Decision Letter:

In reply please quote: NG-A64233-T Straub

19th December 2023

Dear Dr. Straub,

I am delighted to say that your manuscript "Digenic inheritance involving a muscle specific protein kinase and the giant titin protein causes a skeletal muscle myopathy" has been accepted for publication in an upcoming issue of Nature Genetics.

Your paper will be published online after we receive your corrections and will appear in print in the next available issue. You can find out your date of online publication by contacting the Nature Press Office (press@nature.com) after sending your e-proof corrections.

You may wish to make your media relations office aware of your accepted publication, in case they consider it appropriate to organize some internal or external publicity. Once your paper has been scheduled, you will receive an email confirming the publication details. This is normally 3-4 working days in advance of publication. If you need additional notice of the date and time of publication, please let the production team know when you receive the proof of your article to ensure there is sufficient time to coordinate. Further information on our embargo policies can be found here: <https://www.nature.com/authors/policies/embargo.html>

Before your paper is published online, we will be distributing a press release to news organizations worldwide, which may very well include details of your work. We are happy for your institution or funding agency to prepare its own press release, but it must mention the embargo date and Nature Genetics. Our Press Office may contact you closer to the time of publication, but if you or your Press Office have any enquiries in the meantime, please contact press@nature.com.

If you have not already done so, we invite you to upload the step-by-step protocols used in this manuscript to the Protocols Exchange, part of our on-line web resource, natureprotocols.com. If you complete the upload by the time you receive your manuscript proofs, we can insert links in your article

that lead directly to the protocol details. Your protocol will be made freely available upon publication of your paper. By participating in natureprotocols.com, you are enabling researchers to more readily reproduce or adapt the methodology you use. Natureprotocols.com is fully searchable, providing your protocols and paper with increased utility and visibility. Please submit your protocol to <https://protocolexchange.researchsquare.com/>. After entering your nature.com username and password you will need to enter your manuscript number (NG-A64233-T). Further information can be found at <https://www.nature.com/nature-portfolio/editorial-policies/reporting-standards#protocols>

Sincerely,
Kyle

Kyle Vogan, PhD
Senior Editor
Nature Genetics
<https://orcid.org/0000-0001-9565-9665>